# Numerical Fire Spread Simulation Based on Material Pyrolysis—An Application to the CHRISTIFIRE Phase 1 Horizontal Cable Tray Tests

**Tristan Hehnen** [1,2,3]**, Lukas Arnold** [1,2,*] **and Saverio La Mendola** [3]

1  Institute for Advanced Simulation, Forschungszentrum Jülich, 52425 Jülich, Germany;
   t.hehnen@fz-juelich.de
2  Computational Civil Engineering, Bergische Universität Wuppertal, 42119 Wuppertal, Germany
3  Occupational Health and Safety and Environmental Protection Unit, European Organization for Nuclear
   Research, 1211 Geneva 23, Switzerland; saverio.la.mendola@cern.ch
*  Correspondence: l.arnold@fz-juelich.de

**Abstract:** A general procedure is described to generate material parameter sets to simulate fire propagation in horizontal cable tray installations. Cone Calorimeter test data are processed in an inverse modelling approach. Here, parameter sets are generated procedurally and serve as input for simulations conducted with the Fire Dynamics Simulator (FDS). The simulation responses are compared with the experimental data and ranked based on their fitness. The best fitness was found for a test condition of 50 kW/m². Low flux conditions 25 kW/m² and less exhibited difficulties to be accurately simulated. As a validation step, the best parameter sets are then utilised to simulate fire propagation within a horizontal cable tray installation and are compared with experimental data. It is important to note, the inverse modelling process is focused on the Cone Calorimeter and not aware of the actual validation step. Despite this handicap, the general features in the fire development can be reproduced, however not exact. The fire in the tray simulation extinguishes earlier and the total energy release is slightly higher when compared to the experiment. The responses of the material parameter sets are briefly compared with a selection of state of the art procedures.

**Keywords:** CHRISTIFIRE; Fire Dynamics Simulator (FDS); pyrolysis modelling; shuffled complex evolution (SCE); high performance computing (HPC); fire propagation simulation; cone calorimeter simulation; cable tray fire simulation; SPOTPY; PROPTI

## 1. Introduction

In the fire safety engineering community, design fires are a frequently used tool when conducting fire risk assessments. The rigidness of the prescribed fire developments, e.g., hydrocarbon curve, is an obvious limitation. Ideally, the fire development could be simulated, based on the material of the objects involved, as well as ventilation conditions and energy distribution near the fire's location. One way to achieve this goal is to simulate material pyrolysis. Laboratory tests are utilised to support these simulation efforts, as in general the explicit measurement of the material properties is not feasible. There is an implicit hierarchy assumed, in which micro-scale combustion tests, like Thermo-Gravimetrical Analysis (TGA) or Micro-Combustion Calorimetry (MCC), can be used to derive basic parameters to describe the pyrolysis reaction rates, for example, parameters for an Arrhenius equation. Bench-scale tests, like the Cone Calorimeter, can be utilised to determine the thermo-physical parameters. Afterwards, these parameter sets can be used to simulate the fire development in a real-scale setup.

The fire safety engineering community has conducted quite some research within this field, e.g., [1–6]. Rogaume provides an overview over some of the challenges when trying to simulate material pyrolysis in [7], also discussing some optimisation strategies for estimating more complex parameter sets. One of the optimisation strategies is the employment of a shuffled complex evolutionary algorithm (SCE) [8], which is relatively common in fire safety engineering [3,4,9–12] due to its performance [13], and it is also utilised in the work presented here. For other approaches, as much data are taken from experiments as possible and pyrolysis models are built on it [14].

In this contribution, the focus is set on simulating fire propagation in horizontal tray installations, based on pyrolysis of electrical cables. Fire behaviour of cable tray installations has been studied over previous decades. Specifically, the nuclear industry dedicated resources to investigate cable tray fire development on an international level. Some of the more visible projects have been the "Cable Heat Release, Ignition, and Spread in Tray Installations during Fire" (CHRISTIFIRE) Programme [15,16], during the "Propagation d'un incendie pour des scénarios multi-locaux élémentaires" (Fire Propagation in Elementary Multi-room Scenarios—PRISME) [17] sub-projects CORE [18], and CFS [19] were focused on cable fires or the "International Collaborative Project to Evaluate Fire Models for Nuclear Power Plant Applications" (ICFMP) [20]. Research is also conducted in more general terms, like in the "Fire Performance of Electrical Cables" (FIPEC) project [21] initiated by the European Commission. During these projects, different experiments at various scales have been performed, not solely related to cable fires, but also fire and smoke propagation in general. Other experiments were looking into time to failure for cables subjected to fire [22–24], characteristics of different plastic materials used for cable construction [25] or better screening methods for cables to be installed concealed spaces [26], e.g., plenums. It was also determined that the spacing between individual cables or bundles of cables influences the fire propagation [21]. Another aspect of cable fires is their impact on other (safety) systems, for example soot depositions on autocatalytic hydrogen recombiners [27]. A more general review of cable fire phenomena is presented in [28].

Different strategies have been developed to model the fire development in cable tray installations. The performance of several computational fluid dynamics (CFD) codes, to estimate fire development and propagation in cable tray installations, have been compared in the past [29]. Based on Cone Calorimeter data from the FIPEC project mathematical models for material pyrolysis were created [30]. A relatively simple hand calculation model, "Flame Spread over Horizontal Cable Trays" (FLASH-CAT), was developed primarily from data of the CHRISTIFIRE campaign. The FLASH-CAT model was picked up in the frame of the PRISME programme [31] for trays mounted to a wall, where it was implemented into a CFD code (CALIF3S/ISIS) and some parameters were adjusted, such that the model would better recreate the PRISME data. A very similar setup, also from the PRISME programme, was used where the Cone Calorimeter energy release rate was "painted" on a cable tray model, such that each individual surface cell would follow the development of experimental data [32]. The release starts after a certain material temperature of the cable sample was reached, that could be regarded as an ignition temperature. A different approach was followed by Matala and Hostikka [33], where the initial stage was modelled by a design fire. Afterwards the fire was allowed to propagate along the cable trays, based on a material pyrolysis approach.

The described studies above indicate a demand to be able to simulate fire propagation within cable tray installations. The model should be able to take environmental conditions into account that control the fire development with as little prescription, e.g., design fire, as possible. The necessary pyrolysis models exist, but rely on many material parameters that might not easily be acquired. Having said that, to the authors' knowledge, there exist no simulation studies that predict the fire development in horizontal cable tray installations based solely on material parameters. The article presented here proposes an approach to determine effective material parameter sets that allow for the self-consistent fire spread simulation.

In general, a brute force approach is utilised here, for which it is assumed that either only few parameters are known of the studied material, or cannot be transferred directly to the simulation.

Thus, all of the parameters that define a material in FDS are worked upon during the optimisation process. This is specifically the case for the residues, of which no information is available. This work is based upon work that was conducted by Matala et al. [3] and Lautenberger et al. [34]. It follows the concept that material behaviour can be studied sufficiently well in the laboratory scale and, thus, it allows for extrapolation into real-scale scenarios.

The foundation for this work is experimental data obtained by CHRISTIFIRE Phase 1 [15]. Cone Calorimeter tests are chosen as starting point, with a simplified model of the apparatus being utilised in FDS. Employing a numerical optimisation scheme, material parameters are varied in the simplified Cone Calorimeter simulation setup, such as to find the FDS response that is close to the energy release rate data from the experiment. In an inverse modelling process (IMP) the experimental data serves as target, while the material input parameters are adjusted for a simulation response to fit the target. The best parameter set obtained during this process is then utilised in a real-scale cable tray simulation setup. In a validation step, its performance in estimating the fire development is assessed. Furthermore, the results of the presented procedure are compared to selected state of the art prediction approaches.

The outline of this article is given below; sections of the main text are denoted by numbers according to the header numbers while sections of the appendix are denoted by capital letters. At first, a brief overview is provided over the experiments that provide the basis for the conducted simulations Section 2.1. This is followed by a description of the inverse modelling process Section 2.2. A detailed presentation of the different FDS simulation setups is provided, concerning the geometrical Section 2.3.1 and material model of the cable Section 2.3.2, as well as the Micro-Combustion Calorimetry Section 2.3.3, the simplified Cone Calorimeter Section 2.3.4, and tray setups Section 2.3.5. Finally, the reference calculations are introduced Section 2.4, which are afterwards compared to the results of the presented study. Where further information was deemed to be of interest, appendices are referenced. They provide an overview over the parameters and their sampling limits Appendix A, the grid sensitivity of the simulations Appendix B, development of the best parameter sets per generation Appendix C, the Cone Calorimeter Paint methods Appendix D, heat flux assessments in the simulations Appendix E, as well as a brief discussion about different software versions and computers used for the simulations Appendix F.

This work is accompanied by a data repository [35] that contains simulation data, like the input files for FDS and PROPTI, data base files with the IMP results, and the results of the simulations with the obtained material parameter sets. For a brief description, see Section 5.

## 2. Materials and Methods

### 2.1. Experimental Data

The content of CHRISTIFIRE Phase 1 is briefly summarised below. During that experimental campaign, a relatively large number of different cables has been subjected to fire tests of various scales. Thus, data sets of the same cable, but under different conditions, are available. For the procedure presented here, the focus, of which tests to use, was set on MCC, Cone Calorimetry, and horizontal tray installations in the open—the Multiple Tray Tests (MT). As argued below, cable 219 was chosen as the sample, while cables from CHRISTIFIRE Phase 2 have not been considered as of now, however are envisioned to be studied later on.

The choice fell on this specific cable, cable 219, because, in contrast to the other cables:

- It showed good repeatability for the different incident heat fluxes during the Cone Calorimeter tests.
- In the multiple tray tests, the individual trays were completely filled with the cable 219.
- In the multiple tray tests, the cables were neatly arranged to rows that extended over nearly the whole tray width. This allowed the tray representation as a single solid slab in the simulation.

During the Phase 1 of the experimental campaign, MCC tests had been conducted on the individual cable components (insulator and jacket), while using a Pyrolysis Combustion Flow Calorimeter (PCFC) [36]. Samples of about 5 mg from the plastic components were linearly heated up, to 600 °C at a heating rate of 1 °C/s$^{-1}$, within a nitrogen atmosphere. Data were determined, like the specific energy release rate per mass, the mass loss, the amount of solid residue produced, heat of combustion, locations of the maxima of the reaction rates, as well as their respective contributions to the overall decomposition process. This also allows for calculating reaction kinetics parameters, modelled by employing an Arrhenius equation per reaction.

Furthermore, the cables had been subjected to Cone Calorimeter tests. Up to three different, constant radiative heat fluxes were imposed on the samples: 25 kW/m$^2$, 50 kW/m$^2$, and 75 kW/m$^2$. The tests at 25 kW/m$^2$ and 50 kW/m$^2$ were mostly repeated three times, 75 kW/m$^2$ was performed just once. The results of these tests are time dependent data series of the energy release rate (ERR) per unit area.

Afterwards, real-scale tests in horizontal tray installations were performed. Tray racks were placed on scales in a relatively large room, under an exhaust hood. Thus, they were considered as burning in the open, with little influence from the surroundings. From one up to seven ladder-backed trays were mounted above each other. The trays had a width of 0.45 m, a length of 2.4 m, or 3.6 m and they were mounted with a vertical distance of 0.3 m. About 0.2 m below the lowest tray, in the centre, a gas burner was positioned that provided an ignition source of 40 kW ± 5 kW. Energy release rates were determined by means of oxygen consumption in the exhaust stream and by the mass loss rate.

The results from the Tube Furnace and Radiant Panel test are neglected during the presented work.

### 2.2. Inverse Modelling Process

Inverse modelling is used to obtain material parameter sets to describe its behaviour in the simulation. Data obtained from Cone Calorimeter tests serves as target for the inverse modelling process (IMP). An evolutionary, sometimes called genetic, algorithm is utilised in order to carry out the IMP. Specifically, the Shuffled Complex Evolutionary Algorithm from the University of Arizona (SCE-UA), developed by Duan et al. [8] was chosen. It is implemented in the scripting language Python and part of the framework "Statistical Parameter Optimization Tool for Python" SPOTPY [37]. The SCE-UA is used as provided via SPOTPY, without adjustments to the algorithm. The optimisation is conducted over multiple generations. The size of a generation $\Phi$ is determined by Equation (1)

$$\Phi = \left(2n_{\text{parameter}} + 1\right) \cdot n_{\text{complex}},\tag{1}$$

where $n_{\text{parameter}}$ is the number of parameters to optimise and $n_{\text{complex}}$ is the number of complexes within a generation. The number of complexes was chosen to be equal to the number of parameters, which is the default setting of the implementation. In general, it is desirable to reduce the amount of optimisation parameters as the computational complexity, i.e., size of a generation, scales non-linearly with this value.

The root mean square error (RMSE) is calculated between the simulation response and the target data in order to assess the fitness of the different parameter sets. An open-source Python framework, PRPOTI, serves as a communication interface between a simulation software, here FDS, and an optimisation algorithm [38–40].

As stated above, the focus was set on cable 219, in order to streamline the overall process for creating the material parameter sets. In this text, the conducted IMP runs are labelled by indicating the target data (T), the fixed parameters (P), as well as the number of times that sampling limits have been adjusted (L). The following labelling options are utilised:

- Indices for experimental conditions of the target data (T):

  - a: 25 kW/m$^2$
  - b: 50 kW/m$^2$
  - c: 75 kW/m$^2$

- Indices of fixed parameter (P):

  - A: Arrhenius parameters (taken from the report [15])
  - L1: layer thicknesses 2 mm (insulator and jacket)
  - L2: layer thicknesses 4 mm (insulator and jacket)
  - HT: heat of combustion from toluene (`FUEL`)
  - HC: heat of combustion from the report [15]

- Indices of adjusted sampling limits (L):

  - Successively numbered, starting by 0.

  An example of an IMP run label is provided below:

$$T_{a,b,c}P_{A,L1}L_2, \tag{2}$$

where all three irradiance levels are simultaneously used as a joint IMP target, Arrhenius parameters are set to the data from CHRISTIFIRE, the layer thicknesses are set to 2 mm, and it is the second time the parameter sampling limits are adjusted.

The purpose of the different targets is to determine how well the resulting parameter sets represent the Cone Calorimeter experiments.

It was expected that the IMP runs that take multiple irradiance levels into account would yield more robust parameter sets over a wider range of external fluxes. In previous simulations [41] it was realised that specifically the heat flux of 25 kW/m$^2$ was difficult to recreate by FDS. Therefore, some IMP runs contained the 25 kW/m$^2$ case, while others were conducted without it.

### 2.3. FDS Modelling

The foundation for the FDS simulations is a cable model, consisting of a layered surface (`SURF`) of different materials (`MATL`) and released combustible gaseous species (`SPEC`). All of this information is combined to form a simulation setup. In general, the simulation setup can be thought of as being the representation of an experimental setup. In this concept, an experimental setup distinguishes not only between individual apparatuses, like Cone Calorimeter or PCFC, but also their settings, e.g., external heat flux or heating rate. In the following, individual aspects of creating the FDS input are discussed.

#### 2.3.1. Cable Geometry

An electrical cable is an assembly of multiple components, conductors covered by an insulator, wound together and surrounded by a jacket. Each component consists of different materials. In general, the cables themselves cannot be geometrically resolved within the conducted simulations. Therefore, they are represented as a flat obstruction of the flow field (`OBST`). To account for the cable's composition, a layered `SURF` was chosen. The top and bottom layer contain the material model for the jacket, while the insulator is embedded in between. Thus, three layers are used in total to represent the cable. This is considered as necessary trade-off, in an effort to reduce the overall computing times.

In an earlier study [41], a copper layer divided the insulator, which lead to five layers. However, during the inverse modelling process (IMP), the conductor thickness was repeatedly pushed to its lower limit. This behaviour was also described by Matala and Hostikka [42]. Following that example, the conductor material layer was removed, in order to speed up the IMP by reduction of parameters.

### 2.3.2. Chemical Reaction and Material Composition

As basic concept, pyrolysis is understood as the thermal degradation and consumption of a solid, while gas(es) and solid residue(s) are produced. Based on the material temperature, the Arrhenius equation describes the reaction rate [43]. This is basically the mass release rate of a gas from a solid, which can be converted to the energy release rate, if the gas is combustible.

Even if a material appears to be homogeneous on a macroscopic level, it may microscopically consist of a mixture of various components. These components are likely to decompose at different temperature ranges. Micro-scale tests, like MCC, allow for observing such a behaviour. As an example, the plastic material of the cable 219 jacket is assumed to be homogeneous. The experimental data shows two peaks for the cable jacket. This is interpreted as decomposition reactions of two different components, which are represented as two materials (`MATL`) in FDS.

Even though slightly more detailed information on the gaseous species are provided with the report, toluene is used as surrogate fuel. Cables were tested in the Tube Furnace and yields of CO, $CO_2$, HCl, and soot are available. However, in previous simulations [41], heat transfer to the cable surface was identified as a problem in the simulation. This was attributed to the poor spatial resolution of the flame, due to the coarse fluid cells. Therefore, of the implemented species (`SPEC`) in FDS, one was chosen that offers a high radiative fraction for the flame heat radiation—toluene. Its soot yield, $0.178 \ \mathrm{g/g^{-1}}$, was taken from the SFPE Handbook [44].

Both cable components (insulator, jacket) are each allowed to form a solid residue in the simulation. From the experiments, only the mass yield per component for the residues is reported. Parameters, like the (bulk) density, emissivity, thermal conductivity, or specific heat, are not available. Therefore, they have been left to be determined by the optimiser and they are effective parameters. Additionally, they act as buffer material. This means that the optimiser can potentially adjust the residue parameters when it reaches limits for the remaining parameters.

### 2.3.3. Micro-Combustion Calorimetry

The MCC data are not directly part of the inverse modelling. Primarily, it gives an indication of how many pyrolysis reactions are to be expected and are thus modelled. Furthermore, they are used to determine if the Arrhenius parameters, that were found from the experimental data and provided in the report, would emerge out of the IMP. In a further step, the pyrolysis parameters are fixed to the ones obtained from the experiment (called "fixed Arrhenius"), in an attempt to reduce the demand for computational resources.

The simulation setup of the MCC test was conducted by utilising the FDS functionality to run only a TGA analysis with no gas phase simulation, i.e., `TGA_ANALYSIS=.TRUE.`.

### 2.3.4. Simple Cone Calorimeter

For a simple Cone Calorimeter model (SCC), the mesh size is set to an edge length of 47 mm (cube-shaped cells), see Figure 1. For comparison, in earlier work [3] larger cells were utilised (0.1 m edge length). The smaller edge length provides a couple of benefits: it leads to a higher resolution of the gas phase, it fits the size of the retainer frame window that has an edge length of 94 mm and it is close to the target cell size of 50 mm envisioned for the later cable tray simulation setup. With the increased resolution of the flow field, the flame can be more accurately resolved. This, in turn, leads to a better resolution of the radiative heat flux to the sample surface. Furthermore, sample and flame are surrounded by two cells until the mesh boundary is reached and one fluid cell below the sample surface level. This facilitates the formation of a more stable flow field.

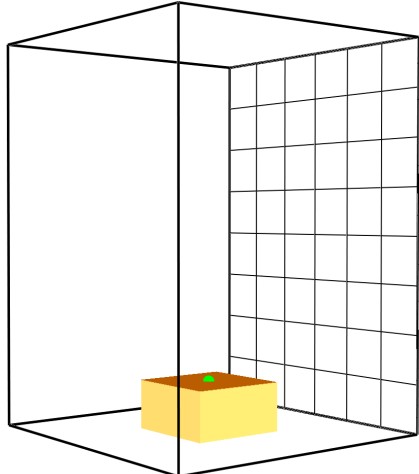

**Figure 1.** Geometrical representation of the improved Simple Cone Calorimeter (SCC) simulation setup in Smokeview. Darker area represents the model of the cable sample. Was also utilised for the Micro-Combustion Calorimetry (MCC) simulations.

2.3.5. Multiple Tray Simulation

During CHRISTIFIRE Phase 1, cable fire behaviour has been tested in horizontal cable tray installations in the open. For this work, Multiple Tray Test 3 (MT3) was chosen, because trays in this test were solely filled with cable 219.

A uniform mesh is chosen for the geometrical representation of the experimental setup of MT3. The cells are cube-shaped, with an edge length of 50 mm, see Figure 2a,b. This choice was mainly driven by an attempt to provide a relatively high resolution, while not being too computationally demanding. It is not based on a mesh sensitivity study. The computational domain is divided into six meshes, as indicated by the differently grey-shaded boxes in Figure 2a,b. The 50 mm cell size is also close to the 47 mm cell size from the SCC simulations, which makes the material parameter sets better transferable to the MT3 simulations. Furthermore, the 50 mm cell size allowed to have five fluid cells between the trays and four between the burner and the lowest tray. In FDS 6.5.3 the principal model for heat transfer within solids is one-dimensional heat conduction in the direction of the surface normal. For this model to take the temperature of the opposite surface into account, when calculating the material temperature, obstructions need to be one cell thick. Thus, the cable layer was created with the thickness of one cell.

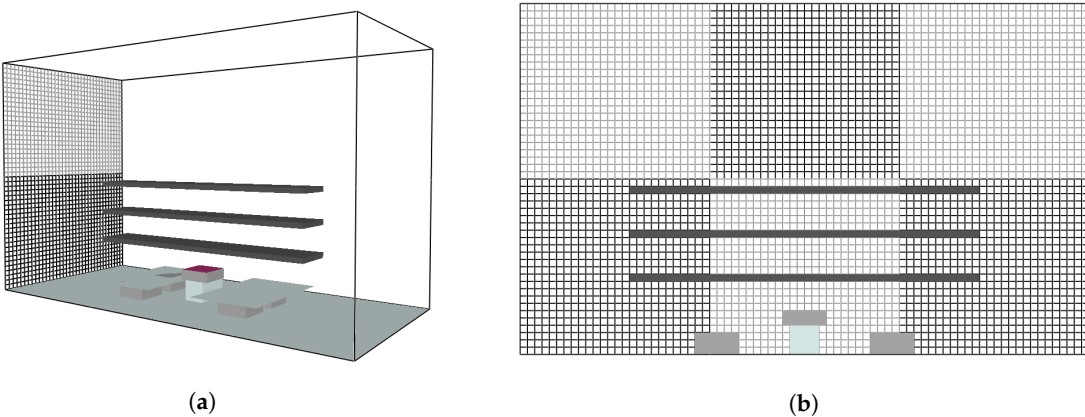

(**a**)                    (**b**)

**Figure 2.** Geometrical representation of the MT3 simulation setup in Smokeview. (**a**) Perspective view. (**b**) Layout of six meshes for the MT3 simulations, indicated by different grey-scale colouring of the meshes.

The metal frame of the trays was neglected during the FDS simulations, because it could not be resolved by the chosen cell size.

In the experiment, the cables were neatly arranged to rows that extended over nearly the whole tray width, see Figure 3. Therefore, it was assumed that the representation of the cables as a single, one cell thick, obstruction would be a reasonable geometrical model. In order to maintain a symmetrical flow field, the width of the tray models is reduced to 0.4 m, thus the burner could be positioned at the centre of the tray rack. The length of the slabs is set to 2.4 m, according to the experimental setup.

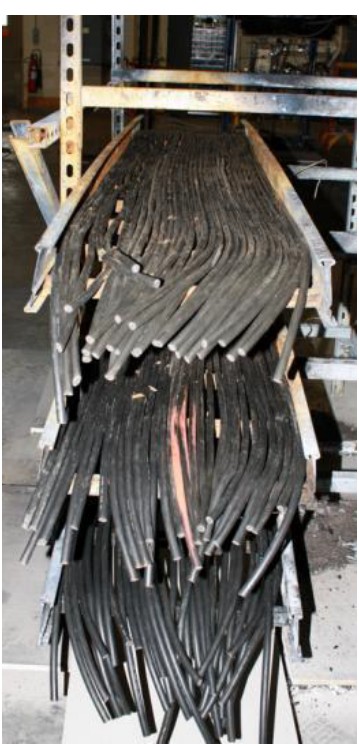

**Figure 3.** Photograph of the cable arrangement in MT3, taken from the CHRISTIFIRE report [15].

The gas burner is modelled by using a boundary condition where toluene is introduced into the computing domain (`VENT`). The same gaseous species, toluene, is used for the gas burner, as well as for the cable material. The burner's energy release rate per unit area is matched to its top surface (`VENT`), such that the it releases 40 kW. The burner starts with the beginning of the simulation and it is shut down after 600 s.

Additionally, the `SURF` that describes the top burner face is assigned a surface temperature that is changed by employing the `TMP_FRONT` parameter and a `RAMP`. Within the first 100 s of the simulation, after a delay of 1 s, the surface temperature of the burner is linearly increased to 410 °C. Afterwards, it is kept constant for 501 s and ramped down linearly for 2599 s. The decrease starts 1 s after the burner is shut off, to somewhat account for a decaying flame, due to small amounts of remaining fuel in the piping between the valve and the burner (in the experiment). The prescribed surface temperature is purely guessed. However, it was deemed necessary to provide some model of the hot burner surface to support the gas flow field that would form, as well as the radiative interaction with the surrounding objects, specifically for the bottom face of the lowest cable tray.

### 2.4. Reference Calculations

To put the results of the IMP in context, three state of the art approaches are followed. This covers an alternative method to determine the model parameters, an approach with prescribed energy release rates, as well as an empirically based model for predicting fire development within a horizontal cable tray installation.

For the first method the Arrhenius parameters are taken from the MCC results, provided in [15]. This is referred to as "fixed Arrhenius" throughout this text, since they are not touched by the IMP in these specific cases. They are used in the same IMP setups, as discussed in Section 2.2, namely the pre-exponential factor $A$ and the activation energy $E$. The reaction order $n$ was set to 1 (FDS default) and the heat of reaction to 1000 kJ/kg$^{-1}$. The remaining parameters are still utilised for the IMP. A further adjustment is to set the layer thicknesses to 2 mm for each, jacket and insulator. This adjustment is a general improvement to the original setup, since the layer thickness and the density thereof are related and it leads to a reduction of necessary simulations to be conducted.

The second method determines parameters, which allow for mapping Cone Calorimeter test results to an object's surface, basically a Cone Calorimeter Paint, see also Appendix D. Different approaches for thermally thick samples are summarised and discussed in ([45] chapter 7). In a recently published paper, this concept was utilised in the context of simulating fire spread in cable tray installations [32]. In the work presented here, Janssens' procedure [45] and the "Beji–Merci procedure" [32] are both used to compare the results from the IMP against. Because no ignition times were reported with the Cone Calorimeter tests [15], they are estimated from the energy release rate data, as described in the Appendix D.

For the third method, the FLASH-CAT model is utilised. It was developed during the CHRISTIFIRE campaign, based on its experimental data [15]. For all multiple tray tests, a calculation was conducted and the respective results provided with the report. Because the model's results for MT3 were already available, they are extracted from the respective plot provided with the experimental data.

*2.5. Section Summary*

The Materials and Methods section provides a brief summary on the experimental campaign and the chosen experiments that are replicated in this study. The structure of the IMP is introduced, as well as the labelling of the different runs. This is followed by a detailed description of the different FDS simulation setups for the simplified Cone Calorimeter and the tray. Specifically, the material composition of the cable model and the decomposition reaction are described. It is concluded by the a presentation of some state of the art reference calculations that are used to compare the results of this study against.

## 3. Results

*3.1. IMP Runs*

3.1.1. Development of the Parameter Sets

Because of the large amount of individual simulations performed during the respective IMP runs, in total more than one million, focus was set to the best parameter sets per generation of the SCE-UA. These parameter sets were then used for different simulations, specifically to assess the performance in the tray setup.

At first, the development of the fitness value of the best parameter set per generation for each IMP run is shown in Figure 4. It is given by the negative root mean square error (RMSE). Each IMP run starts out with a relatively large distance to the target. Within the first 10 to 15 generations, the fitness improves notably and afterwards the rate of improvement decreases.

Appendix A provides some further information on the development of the individual parameters during the IMP.

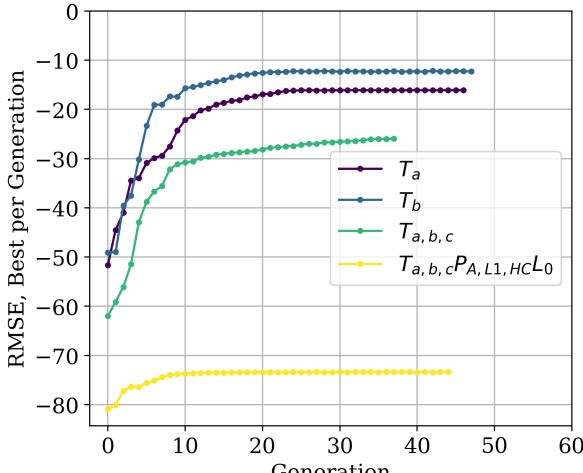

**Figure 4.** Development of the root mean square error (RMSE) values, of the best parameter set per generation, for the inverse modelling process (IMP) runs ($T_*$ and $T_{a,b,c}P_{A,L1,HC}L_0$).

### 3.1.2. Adjusted Parameter Limits

During the IMP runs, some parameters got stuck at their sampling limits. This behaviour is demonstrated in the Appendix A.2, where the parameter development is summarised in ribbon plots. To improve on this, a test series is conducted, in which the respective limits are shifted.

As general procedure, the sampling limits are adjusted by a percentage of the sampling range. In cases where the parameter is stuck at the upper limit, the percentage is added; otherwise, it is subtracted. The percentage is chosen arbitrarily, with a value of about 30%. With these adjusted limits a new IMP run is conducted. The best parameter sets per generation for the adjusted limits are then also run through the whole stack of simulation setups. In general, it can be observed that the shifted sampling limits lead to an improvement of the fitness values. This is demonstrated by groups of the IMP runs $T_bP_{L2,HT}L_*$ and $T_{a,b,c}P_{A,L1,HC}L_*$, see Figure 5.

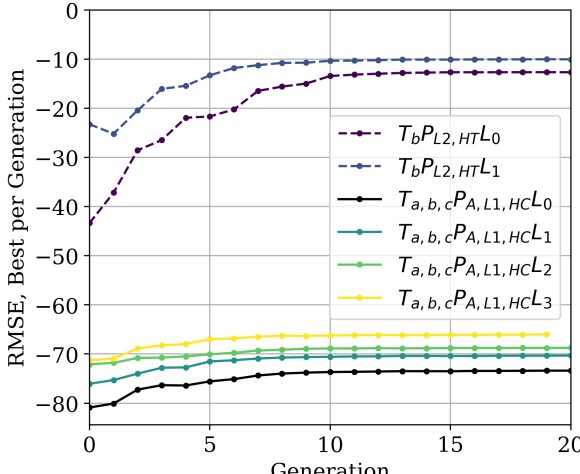

**Figure 5.** RMSE values for IMP runs with adjusted parameter sampling limits. Successive limit adjustments are marked with $L_*$.

### 3.2. Cone Calorimetry Simulation Results

As stated above, not all of the IMP runs were utilising all of the different incident heat flux conditions as target. Despite this, after the conclusion of the IMP runs, all of the best parameter sets were put into simulation setups for all three conditions. This allows for a comparison of the parameter

set's performance under all conditions and specifically compare more rigid (all three tests) to softer (one test) target setups. The following paragraphs describe the results shown in Figure 6.

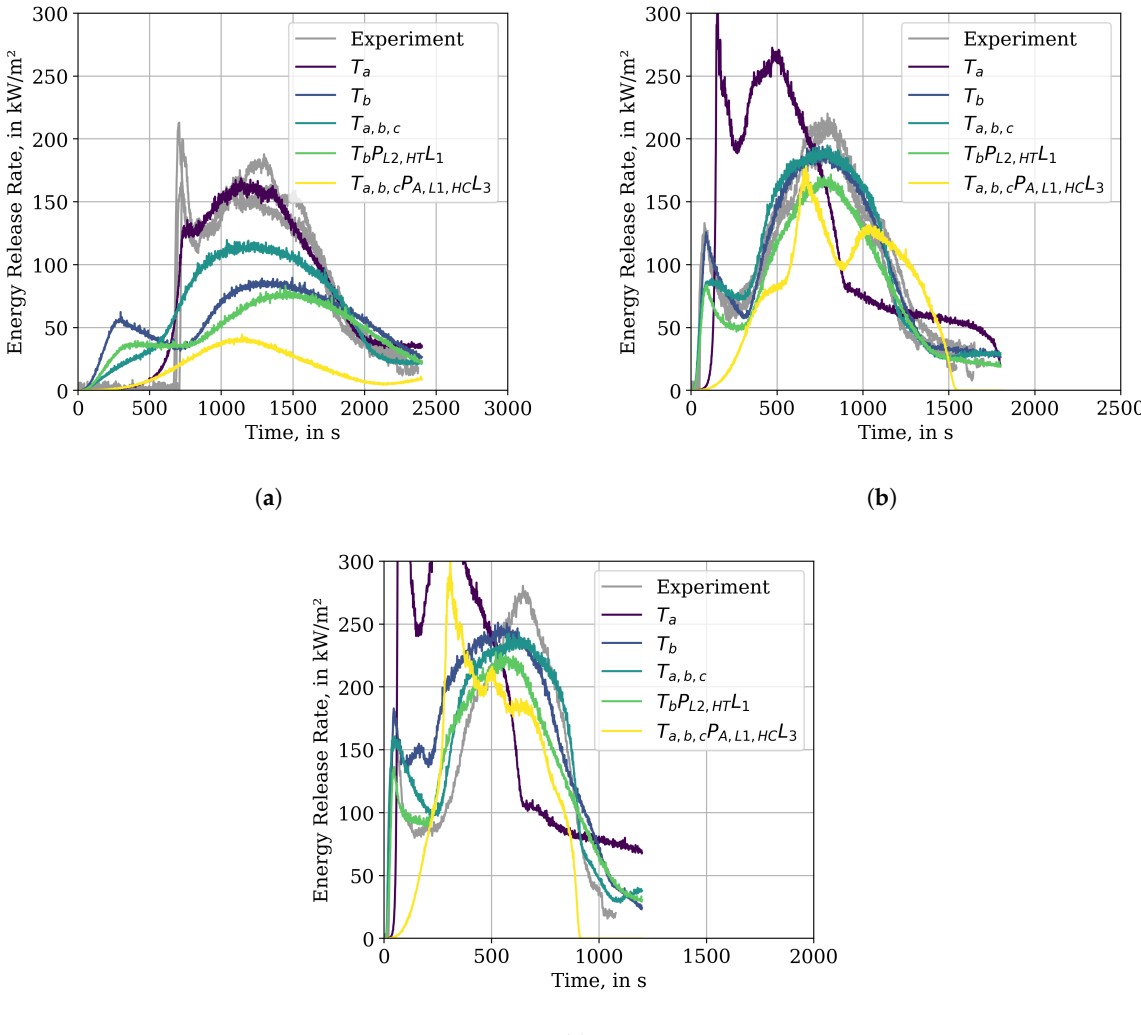

**Figure 6.** Comparison between energy release rates of Cone Calorimeter simulation, across global best parameter sets. (**a**) External heat flux of 25 kW/m$^2$. (**b**) External heat flux of 50 kW/m$^2$. (**c**) External heat flux of 75 kW/m$^2$.

At first, the performance of the IMP results in relation to their respective targets is presented. It can be observed that, for an external radiative flux of 25 kW/m$^2$, only $T_a$ (target 25 kW/m$^2$) is able to represent the experimental data reasonably well, except for the first peak, see Figure 6a. IMP run $T_b$ is able to reproduce the experimental data well for its target of 50 kW/m$^2$, see Figure 6b.

However, comparing the performance of both parameter sets against data that is not used as IMP target shows significant deviation. $T_a$ overestimates the energy release for the other two external flux conditions (50 kW/m$^2$, 75 kW/m$^2$) significantly. Even though the general features of the experimental data can be reproduced. For $T_b$, in the 25 kW/m$^2$ condition, the energy release is over-predicted right from the start, the first peak only reaches a quarter of the energy release as the experiment and happens much earlier. The energy release of the second peak is about a factor of 2 lower as in the experiment. For the 75 kW/m$^2$ data, the first peak is reproduced, as well as the second; however, the intermediate section is over-predicted by a factor of about 1.5.

Visually, $T_{a,b,c}$ shows the best fitness overall three external fluxes, see Figure 6. In the 25 kW/m$^2$ case it is not able to capture the first peak. However, the energy release in the first about 700 s is lower, as compared to $T_b$, and the second peak is represented better. Under the 50 kW/m$^2$ condition, the first peak and valley are reproduced, but appear smoothed. The second peak is represented about as well as in $T_b$. With an external flux of 75 kW/m$^2$, both peaks are reproduced and the valley is captured better as compared to $T_b$, see Figure 6c.

The parameter set of $T_b P_{L2,HT} L_1$ shows slightly better performance in the Cone Calorimeter simulations when compared to $T_b$. Under an external flux of 25 kW/m$^2$ the first peak is less pronounced than that of $T_b$, however the long delay visible in the experiment is also not reproduced. For the 50 kW/m$^2$ case the performance is slightly worse as $T_b$, specifically the first peak is not well captured. In the 75 kW/m$^2$ case, the first peak is still not resolved well, yet the valley is captured more closely than by $T_b$. In all three conditions the peak energy release is less than the the values reported from the experiments.

The state-of-the-art approach, with pre-determined Arrhenius parameters (here based on MCC data), layer thickness, and heat of combustion, is represented by the $T_{a,b,c} P_{A,L1,HC} L_3$ IMP run. Here, all three experiments were used as target. The experimental data of the 25 kW/m$^2$ case could not be reproduced. It shows none of the distinct features and the peak energy release is a factor of about 4 lower than observed in the experiment. A slightly better reproduction of the experiments could be achieved for 50 kW/m$^2$ and 75 kW/m$^2$; however, the energy release rate development diverges significantly.

All IMP runs show difficulties to reproduce the 25 kW/m$^2$ condition. While some show a two-peak-structure, none are able to reproduce the long delay to ignition, which can be seen in the experimental data.

For an overview of the performance of the best parameter sets per generation the reader is directed to the Appendix C, where $T_b$ is provided as an example.

### 3.3. Multiple Tray Simulation Results

A simulation in a MT3 setup was performed for each best parameter set per generation of the IMP runs. The respective simulation results of the energy release rate are plotted and compared to the experimental data that are provided by the report [15].

Of the IMP runs, only $T_b$ is able to reproduce the features of the ERR development, see Figure 7. The first peak, around the time where the burner is switched off at 600 s, is over-predicted by a factor of about 2. After the burner is switched off, the ERR decreases by approximately 40 kW, which is similar to what is observable in the experimental data. In the simulation, the decrease is followed by a peak that overshoots the first peak by about 80 kW, which is again similar to the experimental data, however less pronounced there with about 60 kW. The last peak in the simulation response is a bit lower than the previous peak, while in the experiment the final peak is again about 50 kW larger that the one before. The last two peaks from the simulation overestimate the ERR of the experiment by a third, i.e., about 90 kW. The progression in the simulation is faster when compared to the experiment. At the time, the experiment reaches the second peak, the simulation has reached the third peak and starts to decrease. The different peaks are associated with the propagation of the flame to the next cable layer within the tray installation.

The parameter set of IMP run $T_a$ leads to a massive over-prediction of the ERR and the features could not be reproduced. In contrast to that, the IMP run $T_{a,b,c}$ is not able to cause any significant fire development and does not recover after the burner is switched off. Similar behaviour could be observed for the $T_{a,b,c} P_{A,L1,HC} L_*$ set; here, only the last run, $*L_3$, is shown as example. With the given limits, the parameter sets are not able to achieve fire propagation after the gas burner is shut off.

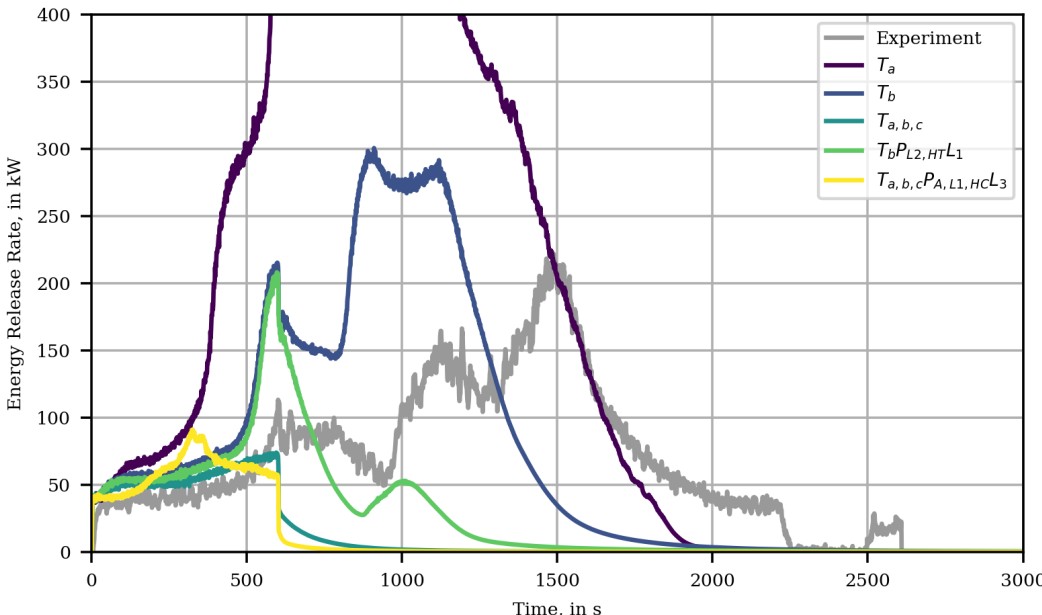

**Figure 7.** MT3 simulation results for a selection of the best parameter sets per IMP run, compared to experimental data. Peak for $T_a$ at about (829.0 s, 856.2 kW).

$T_bP_{L2,HT}L_1$ is able to recover briefly after the burner is shut off, but it does not show meaningful fire development. It is notable that $T_bP_{L2,HT}L_1$ starts from the same trajectory as $T_b$, but looses a lot of its ERR after the burner is cut. Interestingly, an early parameter set of this run is able to reproduce the experimental data in the MT3 setup better, as the best parameter set of the IMP (not shown here).

For the parameter sets that lead to a propagation, it seems that an energy release of about 200 kW needs to be reached to get a sustainable position, which is able to cope with the burner shut-off. Yet, it may not be sufficient in all cases ($T_bP_{L2,HT}L_1$).

### 3.4. Reference Calculation

At first, the results of the FLASH-CAT model are briefly outlined. The model results follow the experimental data; however, it over-predicts the energy release rate, see Figure 8. FLASH-CAT is also not able to resolve the features (peaks) of the experimental data. In contrast to the experiments, where the combustion is sustained up to about 2500 s, the FLASH-CAT model shows a duration of 5400 s. This leads to a higher total energy release than observed during the experiment, a factor of about 2.8. More details on this are provided in Section 3.5.

Simulation responses for the different approaches of the Cone Calorimeter Paint methodology are provided in Figure 8. Across the different variations to the procedures, two clusters of results were generated one with high ERR and one with a low ERR. Because of their similarities, the figure only shows one representative for each cluster. For the low ERR cluster, after the burner is switched off the fire decays until extinction near 2200 s. Only the high ERR cluster shows a significant fire propagation and energy release, where the plot resembles the development of the 50 kW/m² Cone Calorimeter test that served as input for the RAMP. See Appendix D for more details.

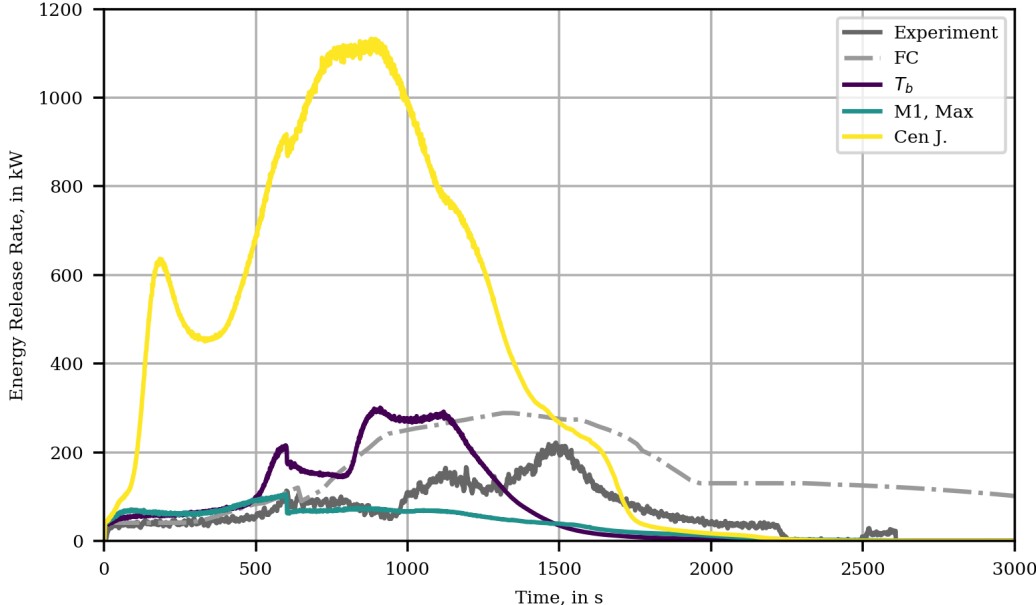

**Figure 8.** MT3 simulation results for different Cone Calorimeter Paint methods and $T_b$ results are provided for comparison. Label "M1" refers to a specific method of the "Beji-Merci" procedure [32], "J." refers to the Janssens' procedure [45].

### 3.5. Total Energy Release in MT3 Setup

The total energy release (TER) for each best parameter set per generation was determined and compared to the experimental value, see Figure 9. Furthermore, data from the FLASH-CAT model (extracted from [15]) and the reference calculations for the MT3 setup are added to the plot. The TER for the experiment is calculated to be about 0.2 GJ, i.e., by integrating over the experimental data series. The FLASH-CAT model shows a significantly higher TER of about 0.5 GJ. Both of the values are provided as constant dashed lines in Figure 9, to allow a comparison for the different model responses.

For the best parameter set of $T_a$ the TER is slightly higher than the value from the FLASH-CAT model. $T_b$ shows a TER that is slightly higher than the value from the experiment. The remaining IMP runs did not perform well during the MT3 simulation, which is indicated by TER values between 0.05 GJ up to 1.2 GJ. For the reference calculations the lowest TER is about 1.0 GJ for the "Beji-Merci" procedure and the highest over 1.2 GJ for Janssens' procedure, overpredicting the TER by a factor of 2 with respect to FLASH-CAT.

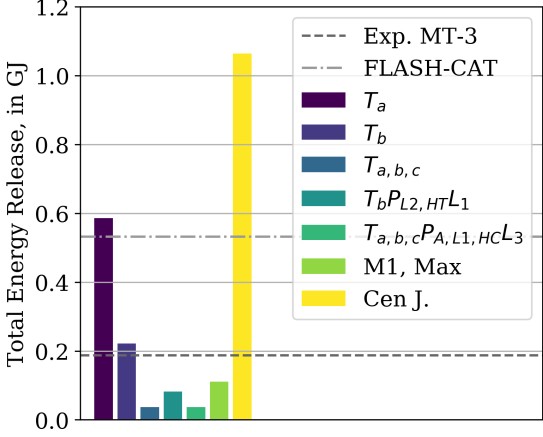

**Figure 9.** Total energy release of the MT3 simulations, as compared with the experimental values and results form the FLASH-CAT (FC) model.

### 3.6. Micro-Combustion Calorimetry Simulations

Even though the results of the MCC experiments were not directly used during the optimisation process, the obtained parameter sets are compared to these data. For each best parameter set, two FDS input files are generated, which contained either the jacket or the insulator material. Utilising the `TGA_ANALYSIS=.TRUE.` functionality of FDS, a MCC simulation is conducted for the best parameter sets per IMP run. The results of the jacket material are presented in Figure 10a, the data of the insulator in Figure 10b. The simulation results are compared to the experimental and model data provided by the report [15]. For the given parameter sampling limits, none of the IMP runs is able to reproduce the experimental data as an emergent phenomenon. Simulations with fixed Arrhenius parameters are not shown here, since they, by construction, produce nearly the same result as the model parameters from the report [15].

Appendix C provides an exemplary overview over the performance of the best parameter sets per generation for IMP run $T_b$ in the MCC simulation setup.

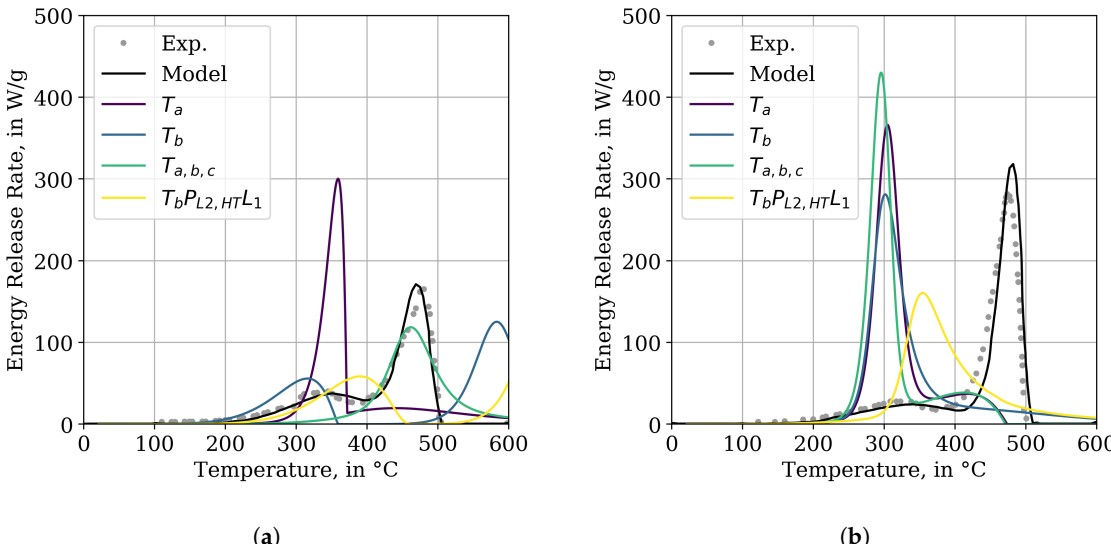

| (a) | (b) |

**Figure 10.** Comparison of energy release rates of MCC simulation (`TGA_ANALYSIS=.TRUE.`) of cable 219, across best parameter sets of different IMP runs, against experimental (Exp.) and model data [15]. (**a**) MCC simulation response of the jacket material. (**b**) MCC simulation response of the insulator material.

### 3.7. Section Summary

The Results section presents the findings of the development of the parameter sets during the IMP runs. The best fit could be achieved for the 50 kW/m$^2$ Cone Calorimeter test ($T_b$). Noteworthy is the IMP run $T_a$, which is the only one to reasonably reproduce the 25 kW/m$^2$ test; however, it massively overpredicts the other tests. In the tray setup, the parameter set of $T_b$ performed best, reproducing the general features that were observed in the experiment. Other parameter set massively overpredict the energy release or cannot recover after the burner is shut off. The tray results are then compared to reference calculations, where different strategies of the Cone Calorimeter Paint lead to significantly different results, either massively overpredicting or under predicting the experimental data. FLASH-CAT overpredicts the energy release rate for nearly the whole experiment, is not able to resolve the peaks that are associated with ignition of the different trays and its runtime is about a factor of 2.8 longer than the experiment. The total energy release for the different methods is compared to the ones from the experiment and FLASH-CAT. Here, $T_b$ is closest to the value from the experiment. The results of the MCC simulations are mentioned here for completeness.

## 4. Discussion

It is obvious that the IMP results presented here do not fit perfectly to the experimental data. Other models, which are strongly based on empirical data, can achieve a better fit, eventually by construction. Yet, their ability to forecast other scenarios is potentially limited. The work presented here is intended to provide a reference to a methodology for finding parameters that reasonably represent the fire behaviour of cable materials. In future work, the parameter transferability and limits of the methodology have to be investigated further.

Still, to the authors knowledge, the work presented here is the first time that a successful material parameter based fire propagation simulation in a cable tray arrangement was achieved.

An important observation is the performance of IMP run $T_{a,b,c}$. Qualitatively, it performs best in the Cone Calorimeter simulations across all external flux conditions, see Figure 6. Though, its fitness values show notably worse performance, as, for example, $T_b$, see Figure 4. On the other hand, the fitness assessment is different due to different target functions. Thus, it is not reasonable to compare fitness values across different IMP. However, in the cable tray simulations, no sustained fire development could be achieved after the burner was switched off. As of yet, we have no explanation for this behaviour. Especially, since the jacket material response in the MCC setup, it is relatively close to the experimental data.

It should be pointed out that the here generated parameter sets are subject to different dependencies. For one, it can be shown that the performance is dependent on the fluid cell size and the solution only converges for higher resolutions, as discussed below. Furthermore, parameter set performance is also sensitive to computer architecture, software versions, and operating systems, see Appendix F.

It is curious that the Cone Calorimeter experiments can be reproduced relatively well, after optimisation. However, the extrapolation to the trays seems difficult. Similar behaviour can be observed when trying to extrapolate from the micro scale to the Cone Calorimeter.

Essential new aspects that have been considered for the overall process, i.e., parameter generation and cable tray simulations, are summarised below:

- The mesh resolution was increased in the optimisation step significantly, when compared with Matala's work [12].
- A wider array of material parameters was taken into account during the optimisation step, including the residues' parameters, and not only the reaction kinetics.
- With toluene, a gas species was chosen, which produces more soot as compared to propane, which leads to a higher radiative fraction of the flame.
- The gas burner was assigned a (highly speculative) surface temperature profile with a slow decay to account for the heat up and feedback during the experiments.

As mentioned above, in order to become a reference for further investigations, an extensive appendix is provided. It contains all of the considered approaches, and summarises the data sets and analysis methods provided in an online repository [35].

### 4.1. IMP

As a general conclusion, it can be demonstrated that the presented approach, using the SCE algorithm and the here formulated constrains, is able to find material parameter sets that are able to reproduce the Cone Calorimeter test responses within FDS 6.5.3, see Figure 6. This is in agreement with findings reported by other researchers, e.g., [4,10,42]. The resulting parameter sets are shown to produce simulation responses that match their individual targets relatively well.

This outcome is not a general statement, as it is the result of the constrains set by the used model (FDS) and the chosen parameter sampling limits. Brief tests conducted to adjust the parameter limits seem to be beneficial to improve the overall fitness, as demonstrated in Figure 5. Despite changes to the sampling ranges, not all parameters could be directed away from the limit they were stuck at, see Appendix A.2. This may be improved by more aggressive changes to the sampling ranges, but also

by taking the improved FDS input (e.g., layer thickness of 4 mm, no `HEAT_OF_COMBUSTION` from the cable material) into account.

The generated parameter sets are to be regarded as effective parameters, in that they are not necessarily realistic values. However, the sampling limits have been chosen to not allow values that are too far away from what could be regarded realistic. Because no information was available for the thermo-physical parameters of the residues of the cable components, they were basically used as buffer material. Due to their parameters being part of the optimisation parameters, the algorithm is able to indirectly influence the material decomposition, by changing e.g., the thermal inertia and the emissivity of the sample.

One could imagine to follow a similar concept with the gaseous species, by introducing a "gaseous buffer". On a simple level, it could mean to mix an inert `SPEC` to the `FUEL`, like nitrogen, and have the algorithm be able to adjust the fraction. However, to cover the initial delay for low flux conditions better, it may be useful to introduce more gas mixtures, e.g., for each cable component, that are associated to the pyrolysis reactions, as discussed by Matala ([3] publication 4). This requires that more detailed information on the composition of the released gas mixture is available. Otherwise, this model would only be as arbitrary as any other. The CHRISTIFIRE Phase 1 report [15] provides information on yields of selected gaseous components from tube furnace tests. Even though this information was not used in the study presented here, to maintain consistency with the selected surrogate fuel of toluene, the argument can certainly be made that the yields alone are not sufficient. Primarily, because they represent the average value during steady-state conditions. This makes it difficult to connect them with the changing temperature profiles present in the other setups, like MCC or Cone Calorimeter. More detailed data-time series would be specifically necessary, when it is to be attempted to connect the mass loss of the sample to the release of gaseous species and the resulting formation of a flame. This is in contrast to the approach followed here, where only the energy release of the flame was considered and the path to the formation of the flame was mostly ignored.

As stated above, the IMP yields good parameter sets for reproducing the Cone Calorimeter results. Thus, it seems that the interaction/relationship of the heat transfer with the pyrolysis processes can be reproduced sufficiently well. The MCC test data are utilised in order to individually check the validity of the pyrolysis process. Yet, the Arrhenius parameters gained deviate from the ones reported from the experiment, despite expected otherwise. To represent the pyrolysis better, one could attempt to use the Arrhenius parameters directly, as obtained from the experiments, i.e., not considered during the optimisation. However, with the fixed Arrhenius parameters, the simulations of the Cone Calorimeter and the multiple tray tests do not yield a reasonable response. This could only be achieved when the IMP was given access to the whole ensemble of the material parameters, therefore globally controlling the interaction. Additionally, it is known that the oxygen concentration in the gas phase around the sample can not be neglected and might significantly influence the material decomposition behaviour [46]. Thus, MCC tests where the sample is heated in a nitrogen atmosphere might not be representative to the conditions during Cone Calorimeter tests. This could also be a cause as to why the generated material parameters are not transferable from the Cone Calorimeter simulations to the MCC simulations.

This leads to the conclusion that these processes and their interactions are not sufficiently well reproduced. It might be caused by incomplete modelling approaches, e.g., the formation of bubbles or cracks, the crude geometrical representation of the cables, i.e., layered, or a combination thereof. This drawback is compensated by producing effective parameters, including the Arrhenius parameters.

Having said the above, it should be pointed out that the `HEAT_OF_COMBUSTION` parameter for the `MATL` was misunderstood, thus it ended up being part of the pool of optimised parameters, originally. The project had progressed too far when this was realised and, due to the computational demand, it was decided to not re-run all IMPs. Thus, the mass fluxes, leaving the solid and entering the gas phase, are not consistent. Yet, the energy release is consistent. Because the optimisation target has been the ERR, this misunderstanding has no direct consequences downstream, i.e., Cone Calorimeter and tray

setup, as long as gained parameter values for the Arrhenius parameter and the `HEAT_OF_COMBUSTION` are used together. This may potentially be the reason why the individual values of the Arrhenius parameters alone are not representing the MCC results.

### 4.2. Gas Phase in Cone Calorimeter Simulations

When compared to previous work [3,41], where the sample surface was resolved with only a single fluid cell, the resolution is increased here. The simplified Cone Calorimeter model has four fluid cells, two by two, to resolve the sample surface instead. This leads to smoother energy release profile, as well as a higher resolution of the flame. Even though the energy release rates seem to converge for higher resolutions, significant grid-dependence can be observed, specifically for the determined heat fluxes towards the sample. The heat flux profiles exhibit a more pronounced development, which follow the profile of the energy release. This is attributed to a higher resolution of the flame and, thus, the improved calculation of the radiative and convective heat fluxes. It highlights the need to take the flame already into account during the IMP as an emergent phenomenon, as opposed to a static prescribed radiative flux. Further information on the grid sensitivity is available in Appendices B and E.1.

### 4.3. Multiple Tray Setup

In general, the observed grid-dependence in the SCC simulations is not expected to have a significantly negative impact on this study. Mainly, because the cell size during the optimisation step is close to the cell size in the MT3 validation setup.

It is further interesting to note that the highest heat fluxes in the tray simulation setups are occasionally about a factor of 2 higher than the imposed conditions in the Cone Calorimeter tests, see Figure A11b. It is not quite clear if this is an "artefact" out of the simulation, or actually observable in real tests, since no data was available to compare this observation to. It also raises the question of whether higher external fluxes in the Cone Calorimeter tests might be necessary to be added to the existing stack of tests. Future (cable) tests could look into this kind of behaviour.

### 4.4. Low Heat Flux Condition

This section discusses the impact of the low heat flux condition, i.e., about 25 kW/m$^2$ and less, in the Cone Calorimeter and cable tray simulation setups. With the given FDS simulation setup, it is very difficult to reproduce the 25 kW/m$^2$ Cone Calorimeter tests, see Figure 6, for example. This is due to the ignition delay observed during the experiments. In IMP run $T_a$ (25 kW/m$^2$ target), the fast increase in the energy release at ignition cannot be reproduced. It is smoothed, i.e., the energy release starts long before the ignition and the first peak is not present.

This primarily seems to be related to how FDS (here version 6.5.3) handles the gas phase combustion. For any given cell, combustion is allowed to occur, even if the fuel concentration is very low. However, in reality, the concentration of fuel gas might not be sufficient for combustion to ensue and just leave the sample surface. This difference leads to a non-zero energy release very early on in the simulation. Additionally, the radiative heat feedback is increased, which, in turn, leads to a faster increase of sample temperatures and, therefore, a quicker release of more combustible gas. As a result, a smoother transition from the pre- to the post-ignition phase is observed, in contrast to the rapid step-like increase that was observed in the experiments.

In this simulation campaign, the released species (SPEC) was only combustible gas, while it is likely, specifically for cables designed to be fire retardant, that inert gaseous species are released first. Thus, it may make it difficult for FDS to deal with the delay visible in the 25 kW/m$^2$ test responses. The approach of Matala, publication 4 in [3], of providing a more detailed decomposition model, which may also release inert gaseous components, could be a solution.

Keeping this shortcoming in the modelling in mind, only the parameters of IMP $T_a$ are able to reproduce the low flux case in the Cone Calorimeter simulation reasonably well. However, applying this parameter set to other, higher, flux conditions in the Cone Calorimeter leads to a significant

over-prediction of the energy release rate and fire development. Similar behaviour can be observed in the cable tray simulation setup, where the fire development is significantly more severe, with respect to a faster progression and a higher peak energy release rate that is about four times higher as in the experiment, see Figure 7.

IMP runs with the remaining experiments as target show difficulties to reproduce the low flux condition. Even though the parameters of the 50 kW/m$^2$ are able to reproduce the 75 kW/m$^2$ behaviour relatively well and vice versa. Under an incident flux of 25 kW/m$^2$ the energy release starts notably earlier with a larger magnitude, as compared to the 25 kW/m$^2$ IMP parameters and experiment.

The importance of the 25 kW/m$^2$ case is not completely clear. On one hand it seems that it does not matter too much when the higher fluxes can be represented well, i.e., by using the higher fluxes as IMP targets. Even though, the energy release rate in the MT3 simulations is notably overestimated around the time of the burner cut-off. On the other hand, assuming a fire propagating along a horizontal fuel bed, every surface element in front of the flame needs to "pass through" a low flux regime to be ignited and differences here should influence the overall speed of the fire propagation. The latter part is demonstrated in Figure A11b in Appendix E.2, where it can be seen that large areas of the cable tray experience low heat flux conditions over the whole course of the simulation.

One way to provide clarification could be to investigate the actual heat flux levels to be expected in large scale configurations during fire experiments. For one to determine whether the observed surface flux levels in FDS are sensible and also to determine if Cone Calorimeter experiments with higher fluxes are necessary in the optimisation process.

*4.5. Geometrical Representation of Cables*

The one-dimensional heat conduction model utilised in the presented simulations could influence the fire development. Specifically, when considering the absence/presence of the copper conductor. This conductor may serve as a heat sink near the fire seat and pre-heating the insulation material further away from the fire [30]. In the Cone Calorimeter simulation setups, this might not be of too much importance, due to the small cable pieces and the relatively uniform heat up of the exposed surface. In that case, it may mainly behave as a heat sink and should be covered by the effective parameter set derived from the IMP. It should be more influential for the cable tray simulations. However, in the given setups it was not possible to resolve the necessary length scales. Investigations of this behaviour may become possible in the near future, due to the three-dimensional heat conduction model added to newer FDS versions, as well as new functionalities allowing for unstructured solids (i.e., `GEOM` name list group).

It has been pointed out, e.g., [21,32], that cables are not necessarily put in a tray, such that they form a continuous slab. Often, they are somewhat loosely packed or combined into bundles. These structures may however be in the sub-grid scale. Matala investigated if individual cables could be modelled in the sub-grid scale, by utilising cylindrical particles [32]. It would be interesting to see how the new unstructured solid method (`GEOM`) method is able to actually resolve the cable models geometrically.

*4.6. Reference Calculations*

As described earlier, the application of both reference methods, Cone Calorimeter Paint and FLASH-CAT, did not yield satisfying results in the investigated setup here. In the following paragraphs, possible reasons are discussed.

For the Cone Calorimeter Paint approach, not only the energy release rate data is needed, but also the thermal parameters ($k\rho c_p$), to determine an effective surface ignition temperature. These parameters were not available for the plastic material, thus density and heat capacity were guessed, based on material properties found on a web page for material properties. This may hamper the comparability between this work and [32], but should be representative for an approach a practitioning fire safety engineer might pursue. In the work presented here, the cables have been modelled as continuous slabs,

instead of "poles", which may be one reason why Beji and Merci could obtain more convincing results in some of the *a posteriori* simulations in [32].

The predicted duration of the fire in the FLASH-CAT approach is based on the combustible mass per tray and a flame front propagation speed [15]. The ignition of the individual trays is controlled as a timed sequence, based on the experimental findings. In contrast to the observation in the experiments and also the simulations presented here ($T_b$); eventually, all trays get involved and are consumed completely. This seems to be the main cause for the much longer duration (Figure 8), as well as the larger magnitude of the total energy release (Figure 9).

*4.7. Robustness of the Model Parameter Sets*

Cable 219 was deliberately chosen to be investigated, because it (a) showed the best reproducibility of the repeated Cone Calorimeter tests, (b) the cable was used in trays containing the same cable, and (c) the individual cables were arranged in rows that made it easier to represent in the simulation.

Other cable tray tests in the campaign showed a more severe fire development, but contained a mixture of various cables per tray and between trays. Having gained confidence that the proposed procedure can generate useful material parameter sets, more cables are to be investigated in future work, e.g., cables 220 and 701.

These two show similar behaviour in the Cone Calorimeter. Furthermore, they were used in tray experiments with mixed cables, where individual trays were filled with a single cable type of 220 or 701, but no mixture of cables within a tray. Thus, it might be easier to reproduce than cable mixtures within trays. It would also make MT8 and MT11 accessible, which showed a much severe fire development, with peak energy release rates of about 800 kW. In the long run, larger scale simulations need to be performed, like the corridors and the vertical shaft setups, from phase 2 of the experimental campaign [16].

With the investigation of further cables, the robustness of the method presented here can be evaluated. More robust material parameter sets allow for the investigation of the influence of parameters, like distances, number of trays, and burner energy release rates and times, cable tray arrangements in corridors can be investigated. This would also set goals for future cable testing campaigns in order to validate the simulation results.

*4.8. Design Proposals for Future Experiments*

For further work on parameter optimisation, to simulate fire propagation, it is important to have access to data from bench-scale, as well as well documented large scale fire experiments. The former is needed during the optimisation, while the latter is necessary to validate the parameter set's performance, which is of specific importance. Simulations that focus on micro- and bench-scale alone, as well as neglecting the gas phase reactions seem not to be sufficient to replicate the large scale fire behaviour. To fill gaps within the existing body of experimental data, future test campaigns should start from the CHRISTIFIRE campaign design as a base line. It is suggested by the authors to focus on one single cable to perform all tests with. Tests in the open offer good cases for simulation software, while presenting a rather modest need for computing resources. However, real world installations are often found in confined spaces, close to walls and ceilings. Therefore, it is necessary to perform similar experiments as the corridor setup presented in phase 2 [16]. Experimental setups with smaller wall sections connected to the trays in the open, similar to FIPEC [21] or like the ones used by Zavaleta et al. [47], could serve as an intermediate step.

Information of peripheral conditions, like material data of surrounding materials, distances to walls (laboratory size/footprint), ventilation conditions, surface temperatures of burner, floor, and other surfaces around the test should also be recorded, to be able to create more comprehensive models.

## 5. Data Repositories

During the course of the IMP runs, and the following analysis of the results, an extensive amount of data were produced. Aiming to allow other researchers to gain a better insight into to whole work, we provide public access to most of the data. Thus, this paper is accompanied by publicly available online data repositories. A summary repository is hosted via Zenodo [35]. It only contains the data that are necessary to reproduce the figures shown here, like the `propti_db.csv` or `*_hrr.csv` files, but not the full simulation results. Additionally, all input files for FDS and PROPTI are provided, as well as the target data.

Furthermore, Jupyter notebooks are provided with the respective repositories. These notebooks are used to process the results from the IMP's and provide an overview by creating various plots. Some are used to guide investigations on the parameter sets, by allowing to create new FDS input files from within the notebooks, as well as presenting the new results within the same notebook afterwards.

From the summary repository at Zenodo, a link will lead to a more comprehensive repository, hosted by the Forschungszentrum Jülich. It contains the full FDS simulations that were created during analysing the IMP results, such that they can be loaded into SmokeView for further study.

**Author Contributions:** Conceptualization: T.H., L.A. and S.L.M.; methodology: T.H., L.A. and S.L.M.; software: T.H. and L.A.; validation: T.H.; formal analysis: T.H.; investigation: T.H.; resources: T.H., L.A. and S.L.M.; data curation: T.H.; writing–original draft preparation: T.H., L.A. and S.L.M.; writing–review and editing: T.H., L.A. and S.L.M.; visualization: T.H.; supervision: L.A. and S.L.M.; project administration: T.H., L.A. and S.L.M.; funding acquisition: T.H. All authors have read and agreed to the published version of the manuscript.

**Funding:** This work has been sponsored by the Wolfgang Gentner Programme of the German Federal Ministry of Education and Research (grant no. 05E15CHA), as part of the CERN Doctoral Student Programme. The authors gratefully acknowledge the computing time granted (project: cjjsc27; application number: 19595) by the JARA-HPC Vergabegremium and VSR commission on the supercomputer JURECA at Forschungszentrum Jülich [48,49].

**Acknowledgments:** The authors would like to thank Kevin McGrattan for fruitful discussions, suggestions and access to experimental data. Furthermore, thanks goes to the members of FCC Fire Collaboration for discussions and suggestions, as well as CERN's Health, Safety and Environmental Protection unit (HSE), for providing access to computing resources.

**Conflicts of Interest:** The authors declare no conflict of interest. The funders had no role in the design of the study; in the collection, analyses, or interpretation of data; in the writing of the manuscript, or in the decision to publish the results.

## Abbreviations

The following abbreviations are used in this manuscript:

| | |
|---|---|
| FDS | Fire Dynamics Simulator |
| SCE | Shuffled complex evolution algorithm |
| TGA | Thermo-Gravimetrical Analysis |
| MCC | Micro-Combustion Calorimetry |
| CHRISTIFIRE | Cable Heat Release, Ignition, and Spread in Tray Installations during Fire experiment campaign |
| U.S.NRC | United States Nuclear Regulatory Commission |
| PRISME | "Propagation d'un incendie pour des scénarios multi-locaux élémentaires" (Fire Propagation in Elementary Multi-room Scenarios) |
| NEA | Nuclear Energy Agency |
| OECD | Organisation for Economic Co-operation and Development |
| ICFMP | International Collaborative Project to Evaluate Fire Models for Nuclear Power Plant Applications |
| GRS | Gesellschaft für Anlagen- und Reaktorsicherheit gGmbH |
| FIPEC | Fire Performance of Electrical Cables |
| CFD | Computational fluid dynamics |

| | |
|---|---|
| FLASH-CAT | Flame Spread over Horizontal Cable Trays |
| CALIF3S/ISIS | Computational fluid dynamics software |
| IMP | Inverse modelling process |
| MT | Multiple Tray Tests |
| PCFC | Pyrolysis Combustion Flow Calorimeter |
| ERR | Energy release rate |
| SPOTPY | Statistical Parameter Optimization Tool for Python |
| PRPOTI | Open-source Python framework that serves as a communication interface between a simulation software and an optimisation algorithm |
| RMSE | root mean square error |
| SCC | Simple Cone Calorimeter simulation setup |
| MT3 | Multiple Tray Test 3 |
| TER | Total energy release |
| FZJ | Forschungszentrum Jülich |
| JURECA | Jülich Research on Exascale Cluster Architectures (supercomputer) |

## Appendix A. Parameter Limits

An overview over the utilised parameters is provided in Tables A1 and A2. These tables also contain experimental data provided, which is limited to the MCC results [15], apart from some guessed values for the reaction orders, heats of reaction, as well as layer thicknesses.

**Table A1.** Overview of the optimisation parameters of the IMP runs for the insulator. Data from the experiment is provided, if available. Layer thickness has been projected from a circular to a rectangular cross section to account for the layered representation in FDS. From the best parameter sets, over all IMP runs, the minimum (IMP Min.) and maximum (IMP Max.) values are provided. The parameter sequence is the same as for the ribbon plots, simply skipping parameters that are not used. (Note: values labelled with "*" are guessed and/or FDS default.)

| "Physical" Parameter | Experiment | IMP Min. | IMP Max. | Unit |
|---|---|---|---|---|
| Insulator Thickness | $1.97 \times 10^{-3}$ * | $3.46 \times 10^{-3}$ | $4.99 \times 10^{-3}$ | m |
| Emissivity | unknown | $3.24 \times 10^{-1}$ | $6.83 \times 10^{-1}$ | - |
| Density | $1.18 \times 10^{3}$ * | $1.02 \times 10^{3}$ | $1.15 \times 10^{3}$ | kg/m$^3$ |
| Conductivity | unknown | $2.05 \times 10^{-1}$ | $2.30 \times 10^{-1}$ | W/(m K) |
| Specific Heat | unknown | $1.49 \times 10^{0}$ | $1.72 \times 10^{0}$ | kJ/(kg K) |
| Heat of Combustion | $3.26 \times 10^{4}$ | $3.89 \times 10^{4}$ | $4.15 \times 10^{4}$ | kJ/kg |
| **Insulator Reaction A** | | | | |
| Pre-exponential factor | $3.99 \times 10^{1}$ | $3.62 \times 10^{1}$ | $4.19 \times 10^{1}$ | 1/s |
| Activation Energy | $4.10 \times 10^{4}$ | $4.24 \times 10^{4}$ | $5.00 \times 10^{4}$ | kJ/kmol |
| Reaction Order | $1.00 \times 10^{0}$ * | $5.02 \times 10^{-1}$ | $2.08 \times 10^{0}$ | - |
| Heat of Reaction | $1.00 \times 10^{3}$ * | $2.54 \times 10^{2}$ | $6.08 \times 10^{2}$ | kJ/kg |
| **Insulator Reaction B** | | | | |
| Pre-exponential factor | $4.50 \times 10^{20}$ | $4.26 \times 10^{20}$ | $5.26 \times 10^{20}$ | 1/s |
| Activation Energy | $3.15 \times 10^{5}$ | $2.36 \times 10^{5}$ | $2.60 \times 10^{5}$ | kJ/kmol |
| Reaction Order | $1.00 \times 10^{0}$ * | $5.07 \times 10^{-1}$ | $3.65 \times 10^{0}$ | - |
| Heat of Reaction | $1.00 \times 10^{3}$ * | $4.84 \times 10^{2}$ | $7.62 \times 10^{2}$ | kJ/kg |
| **Insulator Residue** | | | | |
| Density | unknown | $4.46 \times 10^{2}$ | $6.12 \times 10^{2}$ | kg/m$^3$ |
| Conductivity | unknown | $1.93 \times 10^{-1}$ | $2.00 \times 10^{-1}$ | W/(m K) |
| Specific Heat | unknown | $6.90 \times 10^{-2}$ | $8.60 \times 10^{-1}$ | kJ/(kg K) |
| Emissivity | unknown | $2.98 \times 10^{-1}$ | $6.50 \times 10^{-1}$ | - |

**Table A2.** Overview of the optimisation parameters of the IMP runs for the jacket. Data from the experiment is provided, if available. Layer thickness has been projected from a circular to a rectangular cross section to account for the layered representation in FDS. From the best parameter sets, over all IMP runs, the minimum (IMP Min.) and maximum (IMP Max.) values are provided. The parameter sequence is the same as for the ribbon plots, simply skipping parameters that are not used. (Note: values labelled with "*" are guessed and/or FDS default.)

| "Physical" Parameter | Experiment | IMP Min. | IMP Max. | Unit |
|---|---|---|---|---|
| Jacket Thickness | $3.12 \times 10^{-3}$ * | $3.80 \times 10^{-3}$ | $6.00 \times 10^{-3}$ | m |
| Emissivity | unknown | $3.35 \times 10^{-1}$ | $9.80 \times 10^{-1}$ | - |
| Density | $1.32 \times 10^{3}$ * | $8.55 \times 10^{2}$ | $1.11 \times 10^{3}$ | kg/m$^3$ |
| Conductivity | unknown | $1.73 \times 10^{-1}$ | $2.06 \times 10^{-1}$ | W/(m K) |
| Specific Heat | unknown | $1.29 \times 10^{0}$ | $1.71 \times 10^{0}$ | kJ/(kg K) |
| Heat of Combustion | $2.53 \times 10^{4}$ | $2.31 \times 10^{4}$ | $2.74 \times 10^{4}$ | kJ/kg |
| **Jacket Reaction A** | | | | |
| Pre-exponential factor | $1.51 \times 10^{3}$ | $1.62 \times 10^{3}$ | $2.27 \times 10^{3}$ | 1/s |
| Activation Energy | $5.86 \times 10^{4}$ | $5.50 \times 10^{4}$ | $6.40 \times 10^{4}$ | kJ/kmol |
| Reaction Order | $1.00 \times 10^{0}$ * | $5.16 \times 10^{-1}$ | $2.99 \times 10^{0}$ | - |
| Heat of Reaction | $1.00 \times 10^{3}$ * | $2.52 \times 10^{2}$ | $7.28 \times 10^{2}$ | kJ/kg |
| **Jacket Reaction B** | | | | |
| Pre-exponential factor | $4.92 \times 10^{14}$ | $3.70 \times 10^{14}$ | $4.79 \times 10^{14}$ | 1/s |
| Activation Energy | $2.28 \times 10^{5}$ | $1.94 \times 10^{5}$ | $2.64 \times 10^{5}$ | kJ/kmol |
| Reaction Order | $1.00 \times 10^{0}$ * | $5.15 \times 10^{-1}$ | $2.20 \times 10^{0}$ | - |
| Heat of Reaction | $1.00 \times 10^{3}$ * | $9.20 \times 10^{2}$ | $1.71 \times 10^{3}$ | kJ/kg |
| **Jacket Residue** | | | | |
| Density | unknown | $4.81 \times 10^{2}$ | $5.99 \times 10^{2}$ | kg/m$^3$ |
| Conductivity | unknown | $1.69 \times 10^{-1}$ | $2.29 \times 10^{-1}$ | W/(m K) |
| Specific Heat | unknown | $6.95 \times 10^{-1}$ | $1.03 \times 10^{0}$ | kJ/(kg K) |
| Emissivity | unknown | $3.65 \times 10^{-1}$ | $9.89 \times 10^{-1}$ | - |

*Appendix A.1. Ribbon Plots*

In order to summarise the development of each parameter, over the course of the IMP, ribbon plots were created, see Figure A1. As an example, the jacket layer thickness development of $T_b$ was chosen to illustrate how the ribbon plots are produced. On the left side of Figure A1, a scatter plot provides an overview of each individual parameter value for each repetition (x-axis), within its sampling range (y-axis). The points are plotted with a slight transparency, to indicate where most of them are accumulated. In the centre plot, a histogram is presented that contains the information over the sampling range. This is further compressed, by creating a heat map ribbon of the histogram, shown on the right hand side. Due to the binning necessary for the histogram, all parameter sampling ranges are immediately normalised. Thus, all parameter ribbon plots of a single IMP run can be stacked together horizontally. Furthermore, $y = 0$ then shows the lower limit of the respective sampling range, while $y = 1$ shows the upper limit, as can be seen in the subsequent plots in Figure A2.

With the ribbon plots, the effective development of the parameters during the IMP can be observed. Within the given simulation setups and parameter ranges, some parameters are forced to the limits of the respective sampling ranges. As an example, the ribbon plot of IMP run $T_b$ is shown in Figure A2b.

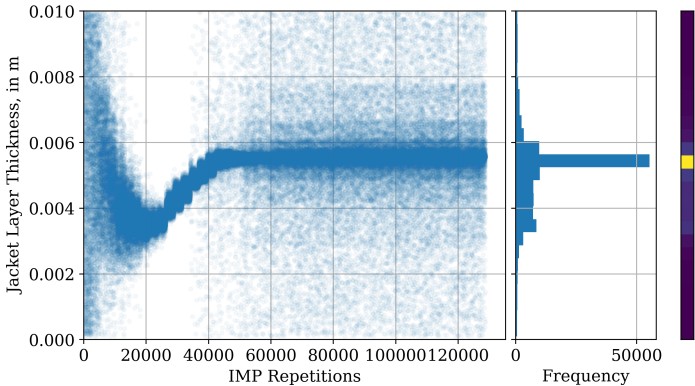

**Figure A1.** Demonstration of how parameter information is condensed. Left side shows the parameter development for the jacket layer thickness during $T_b$. The centre plot shows the frequency of the sampled parameters, distributed over 25 bins. The colour bar at the right side is a heat map ribbon of the histogram in the centre, with yellow being the highest frequency and blue the lowest.

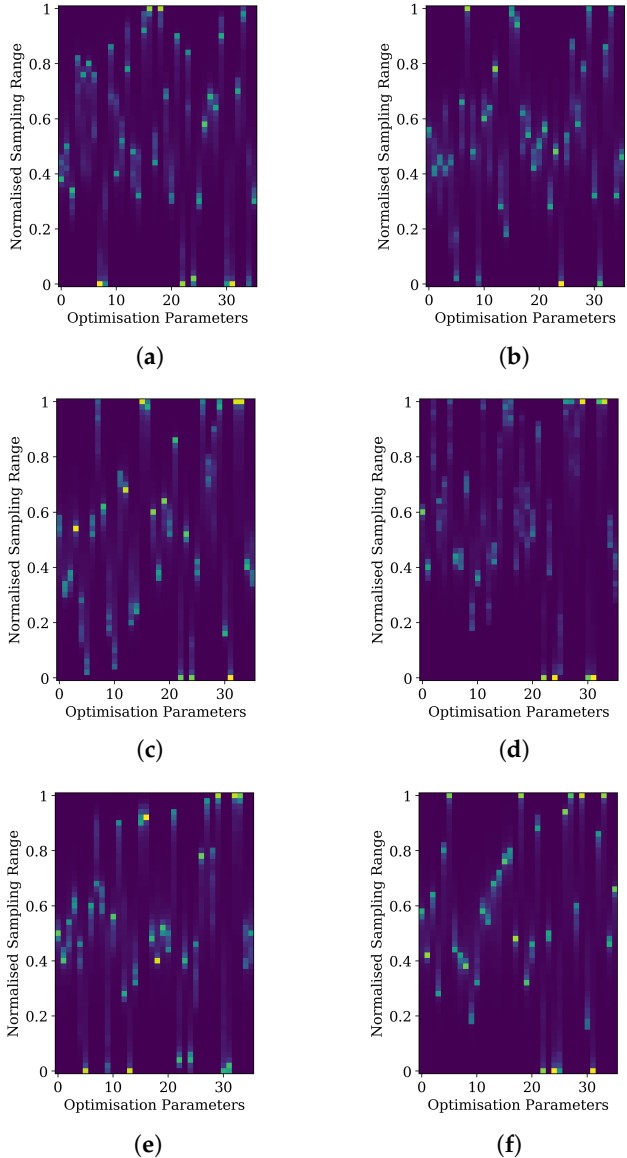

**Figure A2.** Frequency distribution of the optimisation parameters of the different IMP runs ($T_*$). The normalised sampling range is distributed over 51 bins. (**a**) IMP: $T_a$, (**b**) IMP: $T_b$, (**c**) IMP: $T_c$, (**d**) IMP: $T_{a,b,c}$ (**e**) IMP: $T_{b,c}$, (**f**) IMP: $T_{a,c}$.

*Appendix A.2. Shifted Parameter Sampling Ranges*

During the time where the primary IMP runs ($T_*$) were performed, the layer thickness and the density were independently varied. Later on this was changed, because both are related, thus one of them was set to a fixed value. Similarly, the `HEAT_OF_COMBUSTION` was be removed because it is connected to the surrogate fuel concept. For the use case demonstrated in this paper the amount of released fuel is controlled via the Arrhenius model. Therefore, the released energy should correspond to this released fuel and not be scaled by a `HEAT_OF_COMBUSTION` value of a different material.

A more comprehensive assessment on the outcome of shifting the sampling limits is based on IMP $T_{a,b,c}P_A$. The decision, to use IMP runs with fixed Arrhenius parameters for this assessment, was simply made to reduce the computational demand. It is still sufficient to demonstrate the fitness improvement by shifting the sampling limits, see Figure 5.

Figure A3 provides four ribbon plots that demonstrate how the parameter development changes for the adjusted parameter limits. On the left side, Figure A3a, the original IMP run with the fixed Arrhenius parameters is shown. To its right are three successively adjusted runs ($T_{a,b,c}P_AL_1$ to $T_{a,b,c}P_AL_3$). It can be seen that for the given sampling ranges some improvement could be achieved, see also Figure 5. However three parameters, stuck at the upper limit, were not sufficiently influenced.

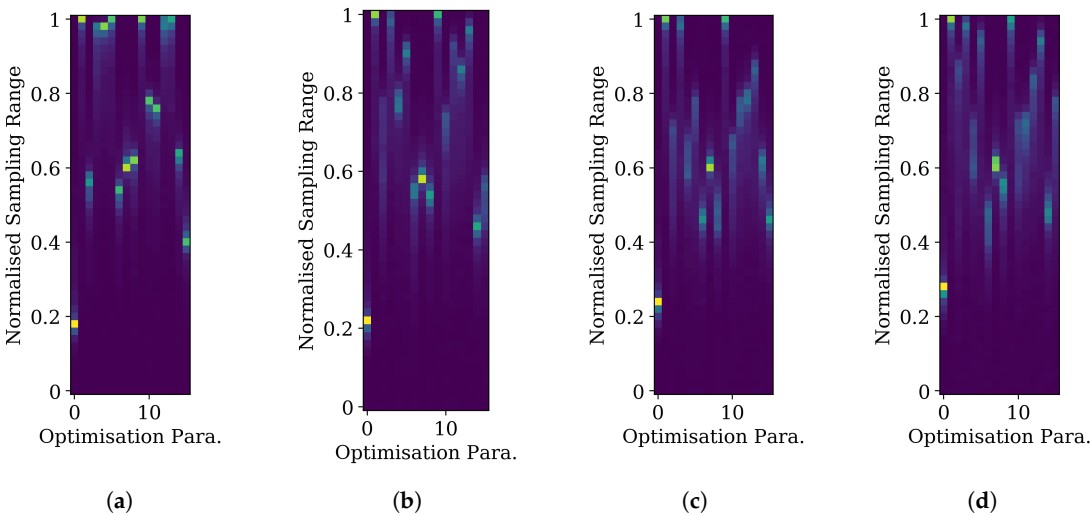

(a)　　　　　　　　　(b)　　　　　　　　　(c)　　　　　　　　　(d)

**Figure A3.** Comparison of the parameter distributions for adjusted parameter limits, with fixed Arrhenius parameters. Sampling ranges have individually been shifted upwards for the stuck parameters, in an effort to improve the overall fitness. Original sampling limits are shown in the left most plot, with successive shifts shown to the right. (**a**) $T_{a,b,c}P_{A,L1,HC}$, (**b**) $T_{a,b,c}P_{A,L1,HC}L_1$, (**c**) $T_{a,b,c}P_{A,L1,HC}L_2$, (**d**) $T_{a,b,c}P_{A,L1,HC}L_3$.

The best parameter sets per generation with the adjusted limits are also tested in the SCC setup. The overall behaviour is quite similar to primary IMP runs, $T_b$ and $T_{a,b,c}P_{A,L1,HC}$. The $T_b$ derivates ($T_{b*}$) fit the 50 kW/m$^2$ quite well, as it is the target, while diverge in similar fashion for the other tests. Responses for the fixed Arrhenius derivates show similar behaviour as IMP $T_{a,b,c}P_{A,L1,HC}$. Comparisons of the simulation responses with the Cone Calorimeter test data is provided in Figure A4.

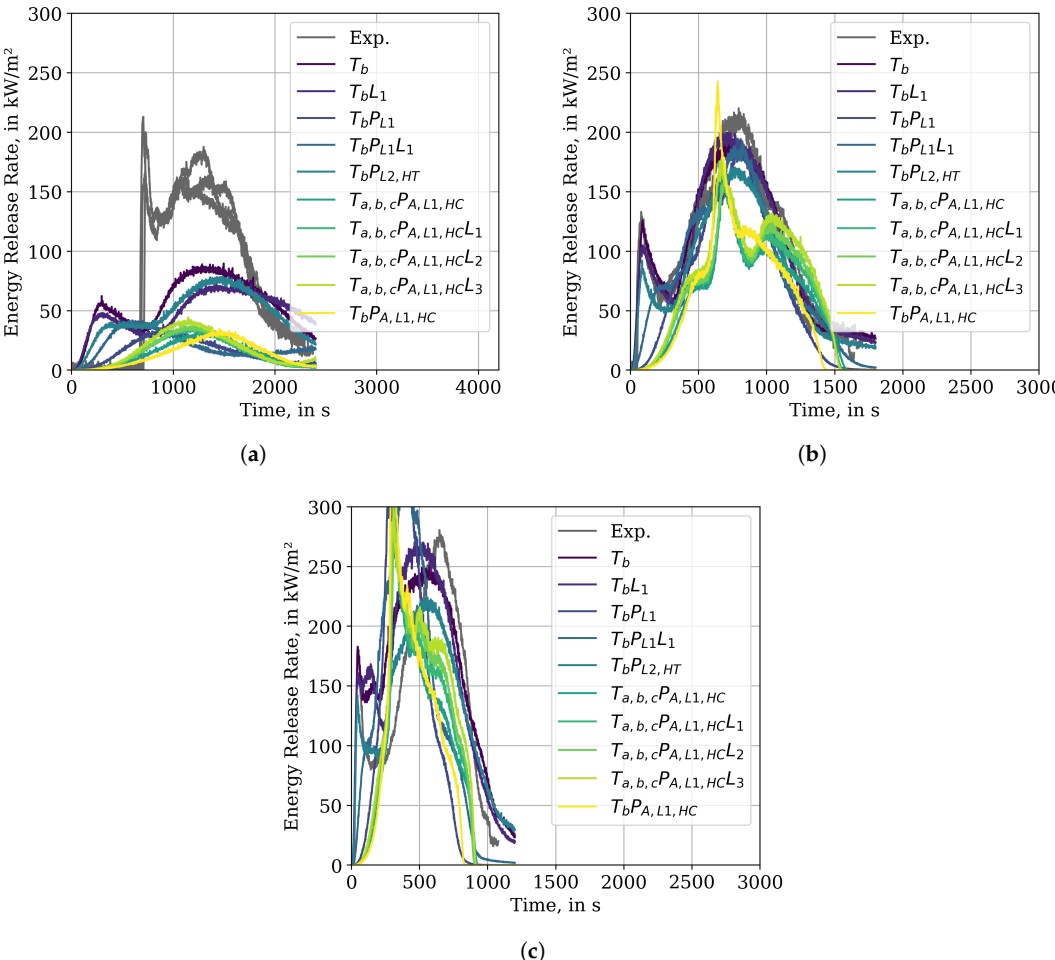

**Figure A4.** Energy release rates of the best parameter set for IMPs with adjusted parameter limits (SCC). (**a**) Incident heat flux of 25 kW/m$^2$. (**b**) Incident heat flux of 50 kW/m$^2$. (**c**) Incident heat flux of 75 kW/m$^2$.

## Appendix B. Grid Sensitivity

*Appendix B.1. Cone Calorimeter*

To investigate the behaviour of the parameter sets for different cell resolutions, the best parameter set of $T_b$ was chosen. The simplified Cone Calorimeter simulation setup SCC was used as base mesh. Its cell resolution was reduced by a factor of 0.5, thus the sample surface was covered by a single cell. To achieve higher resolutions, the cell size of the base mesh was divided by factors 2 to 5. The results are presented in Figure A5 and compared to the experimental data (dashed line). It can be seen that the parameter set does not provide a resolution-independent solution for the energy release rate. For the higher external fluxes it seems to be achievable for a factor of two and higher resolutions. In the 25 kW/m$^2$ setup the convergence is slower. It can also be noted, that a resolution reduction leads to a much more noisy response.

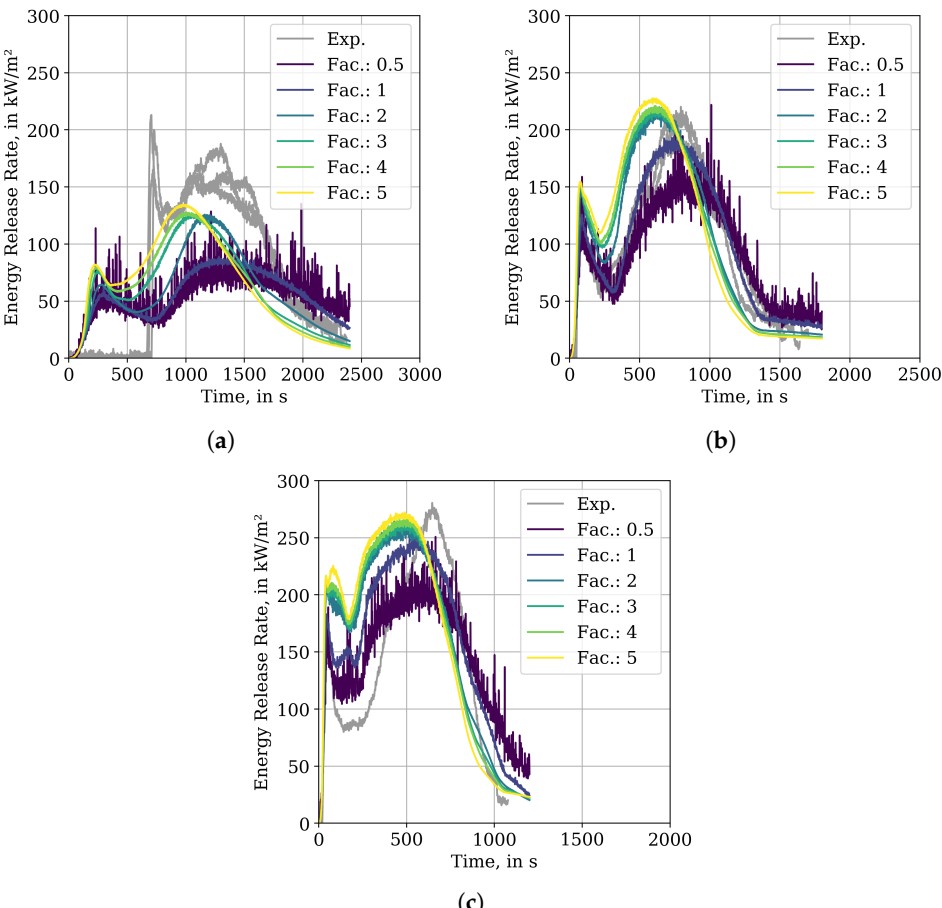

(a)  (b)

(c)

**Figure A5.** Comparison between energy release rates of Cone Calorimeter data, from simulation and experiment (Exp.). Fluid cell sizes were changed by the noted factor, w.r.t SCC (47 mm). The used material parameter set is taken from IMP $T_b$. (**a**) Incident heat flux of 25 kW/m$^2$. (**b**) Incident heat flux of 50 kW/m$^2$. (**c**) Incident heat flux of 75 kW/m$^2$.

*Appendix B.2. MT3*

The best parameter set for IMP run $T_b$ was also utilised in a MT3 simulation setup with 25 mm cells. The results further highlight the strong grid-dependence of the parameter sets, see Figure A6. Note that the fire developed so strongly that the flame was cut at the upper boundary of the MESH and thus the energy release rate is likely higher than the observed maximum value of (958.0 s, 975.3 kW). In the higher resolution case 25 mm, the fire also propagated over nearly the whole tray length, in contrast to the lower resolution. The fire in the lower trays decayed earlier than the fire on the top tray. This lead to the extinguishing of the fire in the top tray before reaching the end.

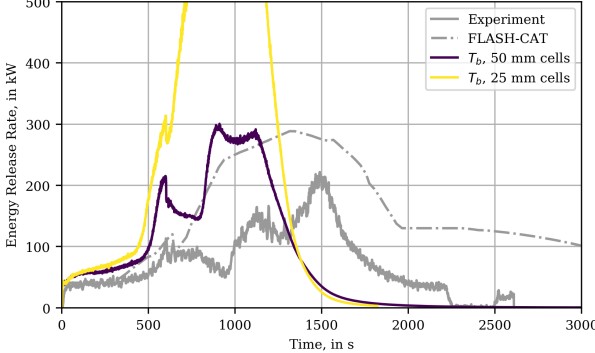

**Figure A6.** MT3 simulation results for the best parameter set of IMP run $T_b$ for with cell sizes of 50 mm (original setup) and 25 mm. The peak of the simulation with 25 mm cells is located at about (958.0 s, 975.3 kW).

## Appendix C. Best Parameter Set per Generation Development

To provide an impression on how the simulation response for the parameter sets change during their development, three plots are provided. They contain the best parameter sets per generation of IMP run $T_b$, applied in the three simulation setups of the stack: MCC in Figure A7a, simple Cone Calorimeter in Figure A7b and MT3 in Figure A7c. For all three plots, in dark blue the results of the first generation are drawn, while the most recent generation is drawn in yellow. For MCC and SCC, the first ten or so generations show some variation and converge afterwards.

Interestingly, despite using the same sequence of parameter sets, the MT3 simulations show much more variation during the parameter evolution, as compared to the SCC simulations.

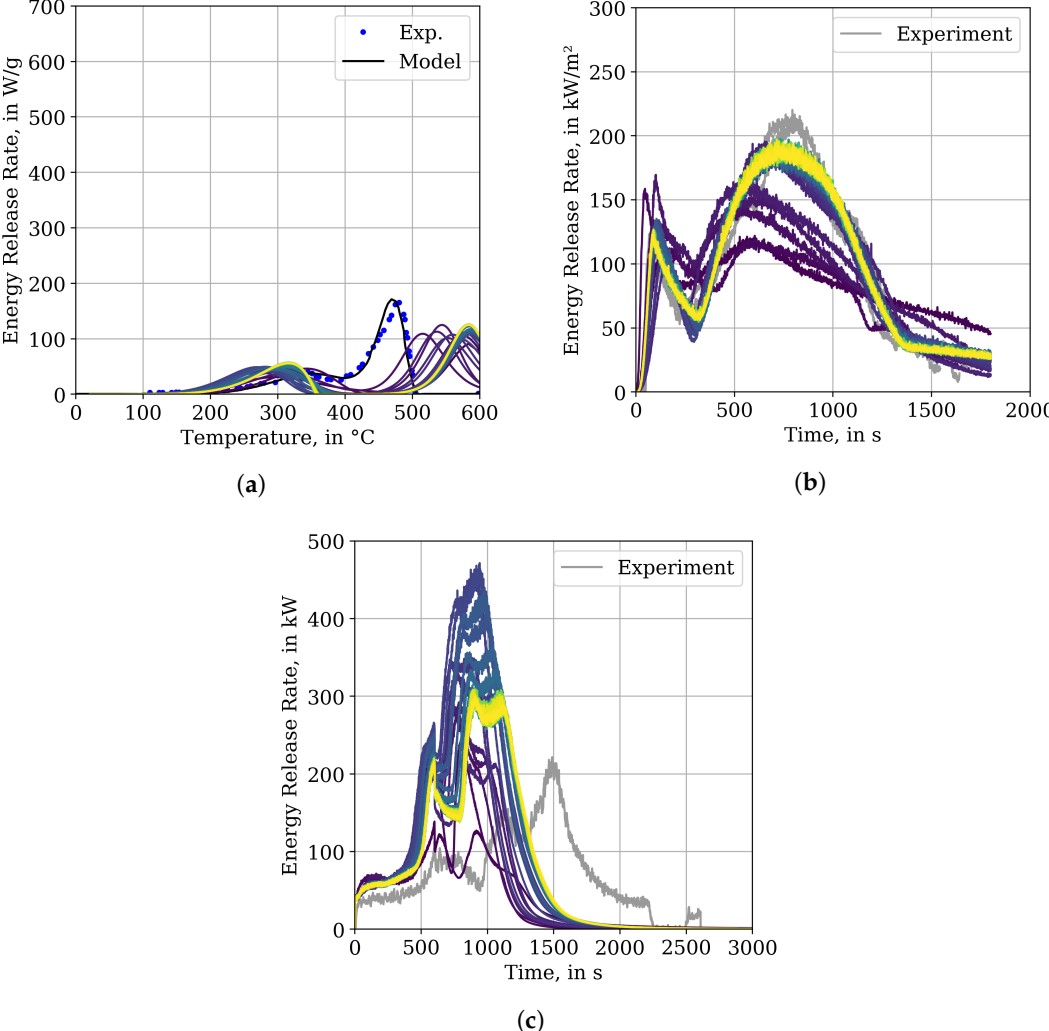

**Figure A7.** Development of the response from the best parameter set per generation, for IMP $T_b$ (50 kW/m$^2$), compared to experimental data [15]. Dark blue represents generation 0, yellow represents generation 47. (**a**) MCC simulation results of the jacket material. (**b**) Cone Calorimeter simulation results. (**c**) MT3 simulation results.

### Appendix D. Cone Calorimeter Paint Methods

Since no ignition times were reported with the Cone Calorimeter tests, they are estimated from the energy release rate data. For each experiment, one ignition time is taken at the beginning of the peak and one roughly in the centre. These times are taken visually from the respective plots. As example see Figure A8a of a repetition of the 50 kW/m$^2$ tests. The blue dot marks the minimum ignition time, while the red dot marks the centre. Furthermore, thermal conductivity (0.1165 W/m$^{-1}$/K$^{-1}$) and density (1175 kg/m$^{-3}$) for chlorosulfonated polyethene, the jacket material, were unknown and taken from a web page providing material data for designers and engineers [50]. To steer the combustible gas release in FDS, a control function (`RAMP`) is created from the first repetition of the 50 kW/m$^2$ data series. The beginning of the data, until reaching the estimated ignition time, is neglected in this control function, with respect to the minimum ignition time. Note: The 25 kW/m$^2$ showed a smoother increase, thus the chosen ignition times are debatable.

For all seven data sets of the experiments, both ignition times, minimum and centre, are determined and are provided in Figure A8b. Each group, red and blue, is used to determine the critical heat flux for Janssens' procedure, by finding the intersection point of a linear fit, through the respective group, with the x-axis. The critical fluxes are marked by a star.

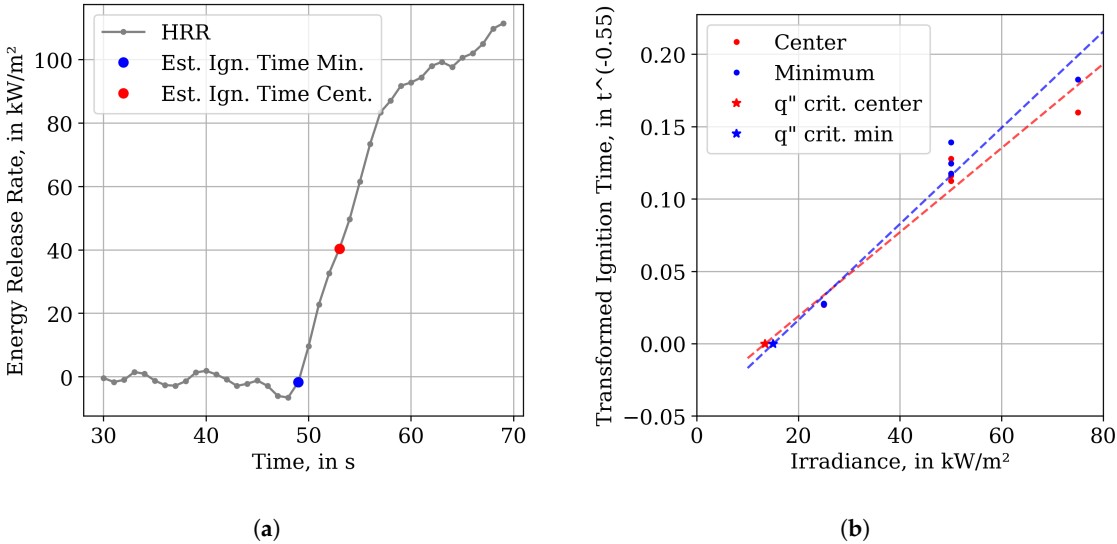

(**a**)                                                                                          (**b**)

**Figure A8.** Illustration of the procedure to estimate ignition times in the Cone Calorimeter experiments. (**a**) Illustration of how the ignition times were estimated, based on a repetition of 50 kW/m$^2$ Cone Calorimeter test. The blue dot indicates the minimum ignition time, while the centre is marked in red. (**b**) Estimation of the critical heat fluxes, based on the "minimum" and "centre ignition times". Note: At 25 kW/m$^2$ six data points are overlapping, three red, three blue.

In the paper [32] three data points were utilised, one for a higher, and two for a lower radiative flux. This lead to two different linear fits drawn into a plot, of which the x-axis intersections were determined. It is not quite clear if, for more available data points for the lower flux, an average value would be desirable or if the lowest and highest values are to be taken into account regardless. It was decided to mimic the previously described procedure. For the irradiance levels of 25 kW/m$^2$ and 50 kW/m$^2$, three data points are available. From each of these clusters the highest and lowest transformed time are taken. From the 75 kW/m$^2$ data point multiple lines are drawn one through each of the highest/lowest points described before. Due to two guessed ignition times from the experimental data, this process is performed for each group resulting in eight linear fits. Some of the lines produce intersection points on the negative side of the x-axis, leading to negative critical heat fluxes. The negative values are ignored and from the remaining positive values the lowest and highest are chosen, mixing both guessed ignition time groups together. Based on these remaining critical fluxes, thermal inertia parameters

are determined following the two methods discussed by Beji and Merci [32]. Thus, four data sets are obtained and MT3 simulations performed.

For the "Beji-Merci procedure", the two methods discussed in their paper [32] followed the basic methodology of Jansssens's procedure. A linear fit is created, taking two data points at different irradiance levels into account. The intersection with the x-axis is used here as critical heat flux as well. Four critical heat fluxes have been determined, one for each estimated ignition temperature and one for heat fluxes. Thus, four data sets are obtained and MT3 simulations performed.

Better values for Janssens' procedure could be produced when using the minimum heat flux where ignition occurs ($q_{min}$), instead of the critical heat flux ($q_{crit}$), as highlighted in the description of said procedure in the Ignition Handbook [45]. However, information about $q_{min}$ was not available in the report [15] and thus only $q_{crit}$ was utilised. A possible approach may be to calculate the mean point between $q_{crit}$ and the lowest irradiance level used in the tests, however this was not attempted here. For more details on the overall procedure, see `ReportConeCalorimeterPaint.ipynb` in [35].

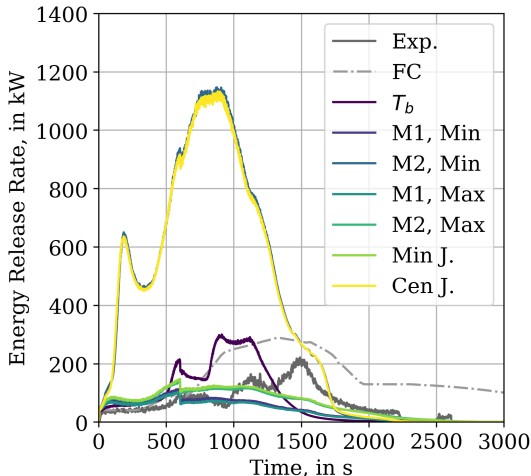

**Figure A9.** MT3 simulation results for different Cone Calorimeter Paint methods. Labels "M1" and "M2" refer to different methods of the Beji-Merci procedure [32], "J." refers to the Janssens' procedure [45].

## Appendix E. Heat Flux Assessment

### Appendix E.1. Cone Calorimeter Radiative Flux Assessment

The fluid mesh resolution study is also used to assess the `INCIDENT_HEAT_FLUX` to the sample surface. The results are presented in Figure A10. A device is employed that integrates the heat flux over the sample surface (`DEVC` with `STATISTICS='SURFACE INTEGRAL'`). The data is smoothed before plotting, by employing a Savitzky-Golay algorithm. A 2nd order polynomial is utilised, with a window length of about 10% of the data points, per data series. Each of the three sub-plots contains a dashed horizontal line to indicate the external flux level that represents the test condition. These values were also defined by the `EXTERNAL_FLUX` parameter in the surface lines (`SURF`) during the IMP runs. It can be seen, as a general observation, that the heat flux at the sample surface in the simulation is always larger than the prescribed external flux value. The heat feedback decreases with lower and increases with higher cell resolution. For the base SCC simulation setup (factor 1), the radiative heat flux from the flame is about 3 kW/m$^2$ higher in the 25 kW/m$^2$ case to up to 10 kW/m$^2$ for the 75 kW/m$^2$ case. For resolutions higher than the base SCC setup, the incident heat flux is significantly higher than the prescribed external flux.

It is interesting to note, that the two peak structure, is mostly not reproduced in the `INCIDENT_HEAT_FLUX` plots for a factor of 0.5. It seems also to smooth out for higher fluxes, see Figure A10c.

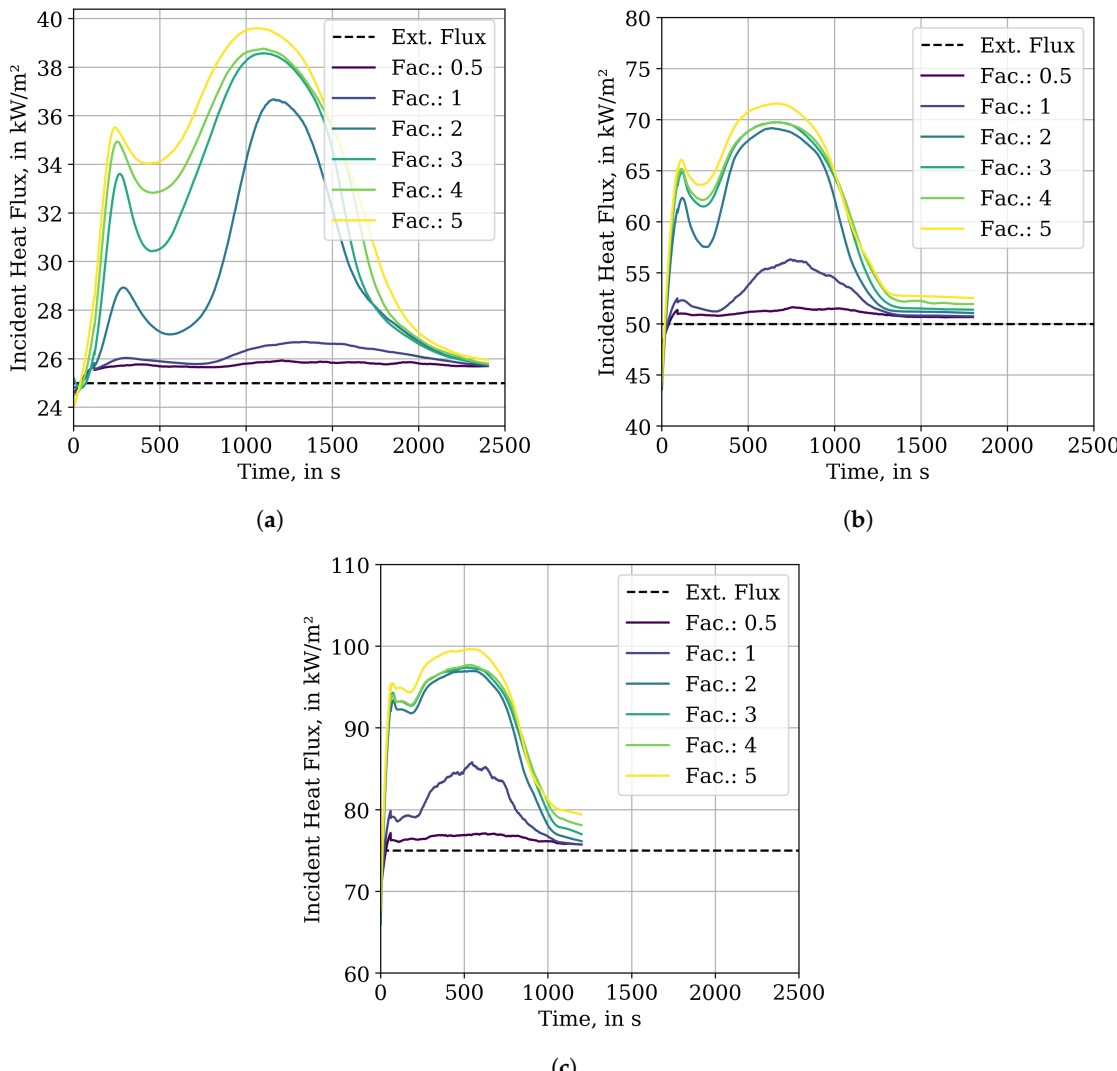

**Figure A10.** Different `INCIDENT_HEAT_FLUX` responses for material parameter set of $T_b$ in Cone Calorimeter simulations with different cell sizes. Cell sizes based on SCC (47 mm) and are changed by dividing through the noted factor. Noise reduction by Savitzky-Golay, 2nd order polynomial, window length about 10% of the amount of data points. (**a**) Experimental condition: 25 kW/m². (**b**) Experimental condition: 50 kW/m². (**c**) Experimental condition: 75 kW/m².

*Appendix E.2. Multiple Tray Simulation Radiative Flux Assessment*

Within the MT simulations, the distribution of the radiative heat flux on the tray surface was tracked, per time step and surface cell. The tracked value in FDS was, among others, the `INCIDENT_HEAT_FLUX`. As an example the `INCIDENT_HEAT_FLUX` distribution for $T_b$ is shown in Figure A11a. Shown are the unfolded surfaces of the three obstructions representing the respective trays. The larger areas of the three groups are the top and bottom faces, with the lower one being at the bottom. Blue colours show a `INCIDENT_HEAT_FLUX` of 0 kW/m² up to 160 kW/m² in yellow.

To assess the general development of the `INCIDENT_HEAT_FLUX` over the whole simulation time and tray surface, a histogram heat map is provided for the best parameter set of $T_b$ see Figure A11b. For each individual time step, a histogram was created for the heat fluxes between 0 kW/m² to 160 kW/m². Three dashed lines show the external heat fluxes that were used during the Cone Calorimeter tests. The heat map colour encodes the surface area receiving a certain radiative flux. During the whole simulation most of the area of the trays only receives low levels of heat radiation, up to about 15 kW/m². At about 400 s, the area increases that receives higher radiative fluxes, which coincides with a growing

fire and its propagation from the bottom of the lowest tray into the space between the lowest and the middle tray, where the bottom of the tray in the middle gets involved. At around 700 s some decay, initialised by the burner cut-off 100 s before, is superimposed by the propagation of the fire into the space between the middle tray and the tray at the top, thereafter reaching the top of the top tray. At about 1000 s the fire starts to extinguish.

The `INCIDENT_HEAT_FLUX` values are demonstrated to reach much higher flux values, as experimental data was available during the IMP, within the simulation. During the full MT3 simulation, individual cells reached significantly higher flux levels as observed in the Cone Calorimeter simulations. They are nearly twice as high, for the resolution of the IMP simulations under the most severe external flux of 75 kW/m$^2$, see Figure A10. For Cone Calorimeter simulations with higher resolutions the peak heat flux of about 100 kW/m$^2$ gets closer to maximum in the tray but is still about 40 kW/m$^2$ short. For now it is not quite clear, if the high flux levels in the trays are an artefact from the simulation or realistic, since no experimental data was available to check it against. The relatively high flux levels in the higher resolutions for the Cone calorimeter seem to point towards higher fluxes are to be expected. They might simply not be able to be reproduced correctly with the low resolution during the optimisation.

Furthermore, as expected, it can be observed that the amount of cells with lower flux levels is relatively large. To visually distinguish areas with trivial heat flux, i.e., zero, the data points are omitted.

An animation of the very similar `GAUGE_HEAT_FLUX` development, as a side-by-side comparison between Figures A11b and A11a, can be found in the `Videos/MT3_GaugeHeatFlux` directory within the data repository [35].

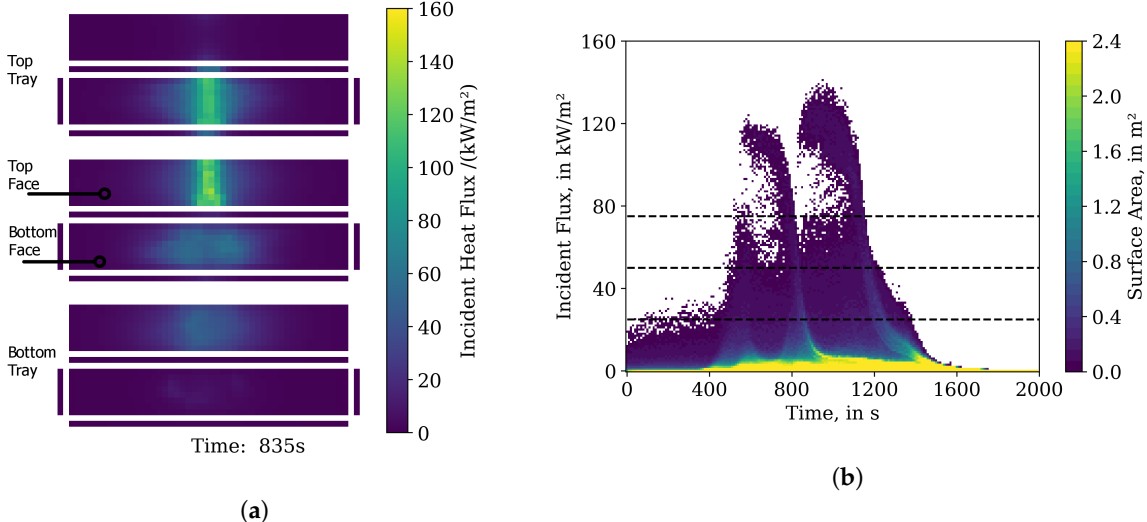

(a)

(b)

**Figure A11.** Simulation results showing the development of the `INCIDENT_HEAT_FLUX` on the cable tray surface for the best material prameter set of IMP $T_b$. (**a**) Unfolded tray surface for a single time step, see also Figure 2a. (**b**) Histogram heat map over the whole tray surface, per time step. External flux for Cone Calorimeter experiments shown as dashed lines. Cells of zero heat flux are omitted (white).

### Appendix F. Computer and Software Versions

A brief assessment of the transfer-ability of the generated parameter sets to different FDS versions, and operating systems, was conducted. Compared are here the FDS versions 6.5.3 [43] and 6.7.0.

Simulations necessary during the IMP were performed on the supercomputer JURECA at the Forschungszentrum Jülich (FZJ) in Germany [48], which utilises an Linux-based operating system. A self-compiled FDS 6.5.3, revision FDS6.5.3-0-gbac6600, was utilised for the IMP, as well as simulations of the best parameter sets per generation after the completion of the respective IMP. This comprised different simulation setups, the MCC, simplified Cone Calorimeter and MT. Thus, consistency between the different setups was ensured. Also on JURECA, FDS 6.7.0, with the revision `FDS6.7.0-0-g5ccea76-HEAD`, was used for comparison with FDS 6.5.3. Both FDS versions are self-compiled against the software libraries available on JURECA. A pre-compiled version of FDS 6.5.3, revision `FDS6.5.3-598-geb56ed1` as provided by NIST via the respective web page, was used on a desktop workstation with a Windows 10 operating system. The FDS input files were the same for all cases, Cone Calorimeter setup and MT3 setup.

The best parameter set of $T_b$ was used as an example for the transferability assessment. Thus, in the subsequent plots FDS 6.5.3 on JURECA is marked with $T_b$. FDS 6.7.0 on JURECA is marked with JU and FDS 6.5.3 on the workstation is marked with WS.

Results of the MCC simulations across the different FDS versions and operating systems show no difference, see Figure A12. For the Cone Calorimeter simulations it can be seen that the peak energy release rates are slightly higher with FDS 6.7.0 (JU) than for both FDS 6.5.3 versions (WS, $T_b$), see Figure A13. The same procedure was followed with the MT3 simulation setup, see Figure A14. In this plot, differences are visible between all FDS versions. Both FDS versions on JURECA show a relatively similar behaviour, with FDS 6.5.3 showing a bit higher peak energy release rate, while the version of the workstation over-predicts the peak energy release by a factor of nearly 2.

This adds a further aspect of dependencies for the model, and highlights that parameter set performance is also sensitive to computer architecture, software versions and operating systems.

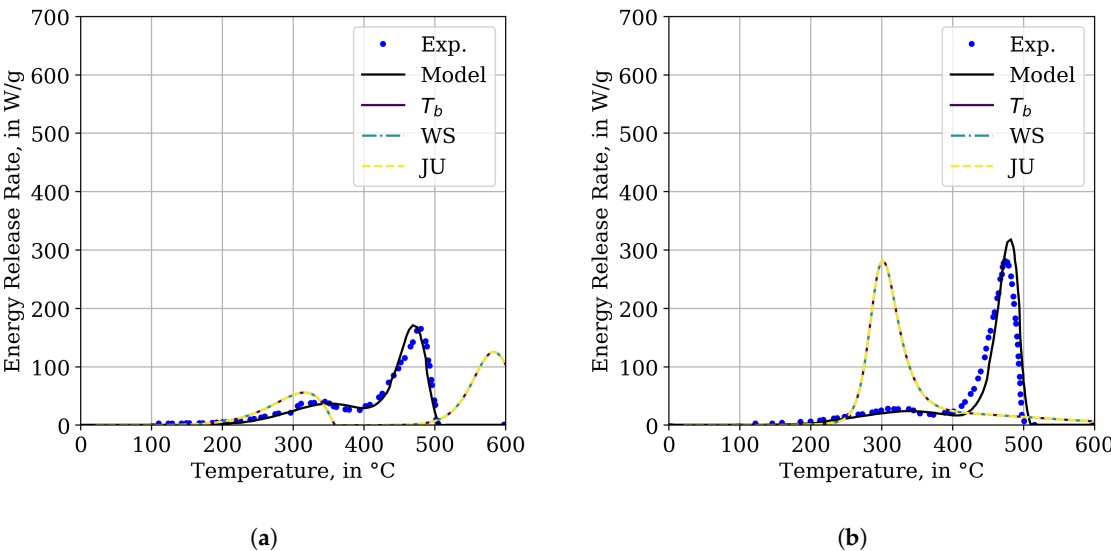

(**a**)                                                                                    (**b**)

**Figure A12.** Comparison between energy release rates of MCC simulation of the best parameter set of IMP $T_b$ (`TGA_ANALYSIS=.TRUE.`). $T_b$: JURECA, Linux, FDS 6.5.3; WS: Workstation, Windows 10, FDS 6.5.3; JU: JURECA, Linux, FDS 6.7.0. (**a**) MCC simulation response of the jacket material. (**b**) MCC simulation response of the insulator material.

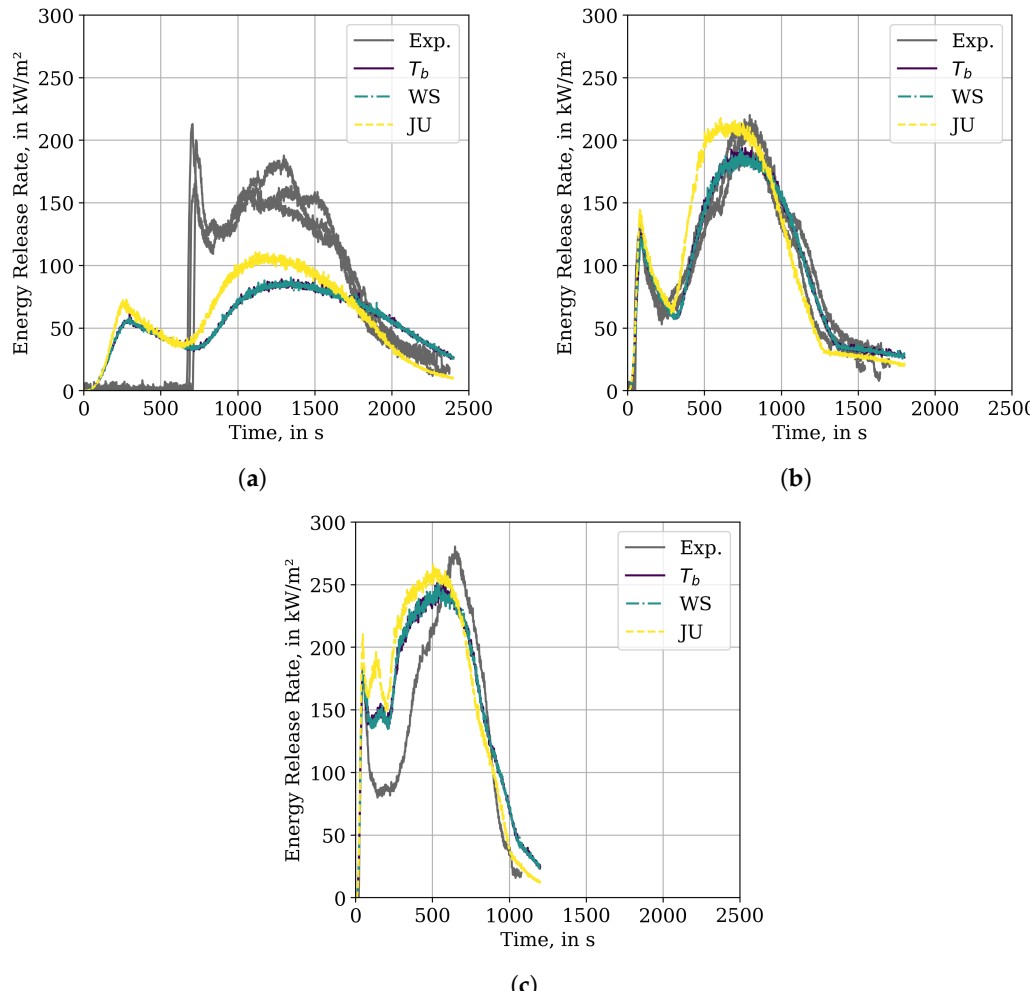

**Figure A13.** Comparison between energy release rates of Cone Calorimeter simulation of the best parameter set of IMP $T_b$. $T_b$: JURECA, Linux, FDS 6.5.3; WS: Workstation, Windows 10, FDS 6.5.3; JU: JURECA, Linux, FDS 6.7.0. (**a**) Experimental condition: 25 kW/m$^2$. (**b**) Experimental condition: 50 kW/m$^2$. (**c**) Experimental condition: 75 kW/m$^2$.

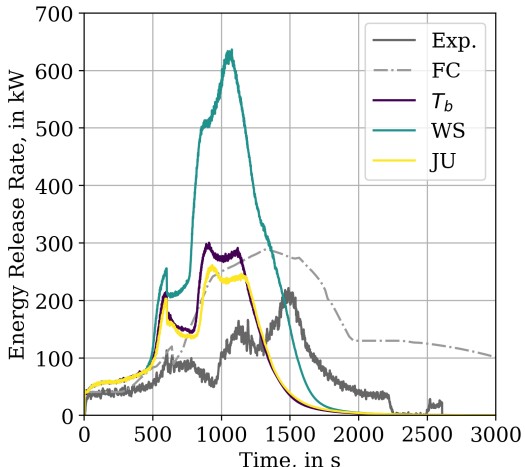

**Figure A14.** Comparison between energy release rates of MT3 simulation of the best parameter set of IMP $T_b$. $T_b$: JURECA, Linux, FDS 6.5.3; WS: Workstation, Windows 10, FDS 6.5.3; JU: JURECA, Linux, FDS 6.7.0.

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
