# Peer review of "Numerical Fire Spread Simulation Based on Material Pyrolysis—An Application to the CHRISTIFIRE Phase 1 Horizontal Cable Tray Tests"

_fire, doi:10.3390/fire3030033_

Round 1
Reviewer 1 Report
The article by Hehnen et al. describes an interesting work focused on the Numerical Fire Spread Simulation of the material pyrolysis. The article is well writing and I consider that it can be accepted after some minor revisions:
1.- Please, the authors can include information about the chemical characterization of the material employed for the simulation, and about the product formed during the pyrolysis in order to understand the chemical reactions, which can affect significantly the activation energy.
Author Response
Dear Reviewer 1,
thank you for the constructive review of our manuscript.
I'm not sure if I understand the question correctly.
In the presented work there is no detailed chemical model of the plastic cable components employed. The material model is primarily a physical one which influences the heat up of the material (density, conductivity, specific heat and emissivity). Furthermore, two products are produced when the cable material is consumed: a solid residue and a combustible gas. An Arrhenius model is employed, that controls the amount of released combustible gas (A, E, n), based on material temperature. The released gas is simply toluene with a soot yield based on tabulated values. The combustion is handled by FDS' default chemical reaction model (simple chemistry in FDS 6.5.3), with toluene being available as a build-in species. The solid residue is again a physical model (density, conductivity, specific heat and emissivity). The amount of solid produced is taken from the MCC test results. The parameter sets that are determined by the IMP are regarded as "effective", or "artificial", that are able to show a similar behaviour in the simulation as was observed in the experiments. Due to their effective nature, the parameter sets can also only be used together. I doubt that focus on the activation energy alone - or any other individual parameter for that matter - provides any meaningful information.
Could you please rephrase your question or does the above explanation provide the information you were looking for?
Kind regards,
Tristan Hehnen
Reviewer 2 Report
Comments:
- The subject matter of the author's investigation is very interesting. However, the surveying of the existing literature and the organizations of their research are not good at all. So, in the first place, in the existing introduction, the related and existing research works which are available in the open the literature MUST be surveyed in an extensive manner to figure out the relevant and clear ''research gap''. Once the ''research gap'' (i.e., the objective of this investigation) is revealed clearly, then the rest of the parts of this manuscript MUST be prepared in such a way that every part seems to supply the answers to fill the '' research gap''. This is one of the biggest drawbacks of this manuscript. Thus, a major revision is recommended.
- Since this manuscript is too long, it is suggested to provide a summary after each section to connect the next section smoothly. It is a MUST issue to go through this manuscript by the potential readers smoothly and easily. This even it is absent in the current version, and it is another big drawback. So, a major revision is necessary.
- Too much data are presented in this manuscript, which makes the barriers in going through this manuscript easily. So, it is recommended to consider only essential data to make a good story of this manuscript. If it is difficult, then this manuscript can be split into two papers.
- It is also suggested the authors to present their revised version to us by highlighting revised parts only different color. Consequently, it would be easier to check what kind of modification has been done by the authors in their manuscript.
In summary, I am suggesting to authors to carry out a major revision of their manuscript in order to qualify the criterion for acceptance in the Journal of Fire in MDPI.
Author Response
Dear Reviewer 2,
thank you for the constructive review of our manuscript. We have addressed all issues that were pointed out. The changes are highlighted in the new version of the manuscript.
"1. The subject matter of the author's investigation is very interesting. However, the surveying of the existing literature and the organizations of their research are not good at all. So, in the first place, in the existing introduction, the related and existing research works which are available in the open the literature MUST be surveyed in an extensive manner to figure out the relevant and clear ''research gap''. Once the ''research gap'' (i.e., the objective of this investigation) is revealed clearly, then the rest of the parts of this manuscript MUST be prepared in such a way that every part seems to supply the answers to fill the '' research gap''. This is one of the biggest drawbacks of this manuscript. Thus, a major revision is recommended.":
- We have expanded on the literature in the introduction (lines 42 - 44, 48 - 54 and 67 - 69).
- However, I have the slight suspicion that I might have overlooked something obvious, due to the way the point is expressed. Having cited FIPEC, PRISME and specifically CHRISTIFIRE, with works of other researchers trying to reproduce the experiments by means of simulation I was relatively confident that the most relevant literature, with respect to the presented study here, was mentioned. Could you please point us to work you think we might have missed that is relevant to the presented work?
- We have adjusted the introduction to better formulate the research gap and goal of the work (lines 70 - 84).
"2. Since this manuscript is too long, it is suggested to provide a summary after each section to connect the next section smoothly. It is a MUST issue to go through this manuscript by the potential readers smoothly and easily. This even it is absent in the current version, and it is another big drawback. So, a major revision is necessary.":
- We have added summaries to the ends of section 2 (lines 317 - 323) and section 3 (lines 460 - 472).
"3. Too much data are presented in this manuscript, which makes the barriers in going through this manuscript easily. So, it is recommended to consider only essential data to make a good story of this manuscript. If it is difficult, then this manuscript can be split into two papers.":
- We are aware that a lot of information is presented here. In an attempt to provide a better reading experience, we had already moved parts of the information from the main text to the appendix. We decided to follow this approach, because we firmly believe that the information is still useful to be kept in close vicinity to the main text, instead of, for example, in the data repository alone. Specifically, because the appendix answers frequently asked questions right away, e.g. the *always* raised question on grid sensitivity or parameter values, as well as it provides means to present interesting observations, like the influence of software versions and computer architecture on the simulation results, which otherwise would be lost.
"4. It is also suggested the authors to present their revised version to us by highlighting revised parts only different color. Consequently, it would be easier to check what kind of modification has been done by the authors in their manuscript.":
- All additions and changes are marked in red within the document.
Kind regards,
Tristan Hehnen
Round 2
Reviewer 1 Report
I consider that the manuscript can be accepted in the present form
Reviewer 2 Report
Comment:
The authors have revised their manuscript as suggested, and consequently, the quality of the revised version is improved. Thus, this revised version can be accepted for the publication in the Journal of Fire subjected to the addition of a section named ''conclusion'' before ''the direction of future work''. Besides, the section 5 must be adjusted before the ''conclusion'', and ''the direction of future work'' should be the last section before the section of appendices.